EMBO
Molecular Medicine

# Reinstating plasticity and memory in a tauopathy mouse model with an acetyltransferase activator

Snehajyoti Chatterjee[1,2,†] iD, Raphaelle Cassel[1,2,†], Anne Schneider-Anthony[1,2,†], Karine Merienne[1,2], Brigitte Cosquer[1,2], Laura Tzeplaeff[1,2], Sarmistha Halder Sinha[3], Manoj Kumar[3] iD, Piyush Chaturbedy[4], Muthusamy Eswaramoorthy[4], Stéphanie Le Gras[5], Céline Keime[5], Olivier Bousiges[1,6], Patrick Dutar[7], Petnoi Petsophonsakul[8], Claire Rampon[8], Jean-Christophe Cassel[1,2], Luc Buée[9], David Blum[9] iD, Tapas K Kundu[3,*] iD & Anne-Laurence Boutillier[1,2,**] iD

## Abstract

Chromatin acetylation, a critical regulator of synaptic plasticity and memory processes, is thought to be altered in neurodegenerative diseases. Here, we demonstrate that spatial memory and plasticity (LTD, dendritic spine formation) deficits can be restored in a mouse model of tauopathy following treatment with CSP-TTK21, a small-molecule activator of CBP/p300 histone acetyltransferases (HAT). At the transcriptional level, CSP-TTK21 re-established half of the hippocampal transcriptome in learning mice, likely through increased expression of neuronal activity genes and memory enhancers. At the epigenomic level, the hippocampus of tauopathic mice showed a significant decrease in H2B but not H3K27 acetylation levels, both marks co-localizing at TSS and CBP enhancers. Importantly, CSP-TTK21 treatment increased H2B acetylation levels at decreased peaks, CBP enhancers, and TSS, including genes associated with plasticity and neuronal functions, overall providing a 95% rescue of the H2B acetylome in tauopathic mice. This study is the first to provide *in vivo* proof-of-concept evidence that CBP/p300 HAT activation efficiently reverses epigenetic, transcriptional, synaptic plasticity, and behavioral deficits associated with Alzheimer's disease lesions in mice.

**Keywords** acetylation; Alzheimer's disease; CREB-binding protein; learning; transcription

**Subject Categories** Neuroscience; Pharmacology & Drug Discovery

## Introduction

Alzheimer's disease (AD) is characterized by the accumulation of amyloid beta peptides and abnormally phosphorylated Tau proteins, and a progressive impairment of plasticity and memory functions, ending in massive neuronal loss and dementia (Serrano-Pozo *et al*, 2011). Available treatments have minimal or no effect on the course of the disease, and most advanced strategies targeting the pathological hallmarks of AD (e.g., Aβ immunotherapies) have been recently proven unsuccessful (Abbott & Dolgin, 2016; The Lancet, 2017). Alternative strategies aim to restore neural circuits and plasticity (Canter *et al*, 2016). Epigenetic changes have emerged as important contributors of neurodegenerative diseases (Schneider *et al*, 2013; Coppede, 2014; Francelle *et al*, 2017), including AD (Fischer, 2014; Bennett *et al*, 2015). Histone acetylation has been particularly studied since it is an important regulator of plasticity and memory formation (Peixoto & Abel, 2013; Zovkic *et al*, 2013). Altered acetylation regulations are thus presumably involved in cognitive deficits. For instance, increased histone deacetylase 2 (HDAC2) levels leading to decreased histone acetylation have been observed at genes that are important for learning and memory, both in mouse models of AD and in post-mortem brains from patients with early-stage AD (Graff *et al*, 2012). Hence, several histone deacetylase (HDAC) inhibitors have been tested in different AD mouse models and some showed significant improvement in plasticity and memory functions (reviewed in Fischer, 2014; Graff & Tsai, 2013). Yet, few genome-wide-scale approaches have been performed using

1   Laboratoire de Neuroscience Cognitives et Adaptatives (LNCA), Université de Strasbourg, Strasbourg, France
2   LNCA, CNRS UMR 7364, Strasbourg, France
3   Transcription and Disease Laboratory, Molecular Biology and Genetics Unit, Jawaharlal Nehru Centre for Advanced Scientific Research, Bangalore, India
4   Chemistry and Physics of Materials Unit, Jawaharlal Nehru Centre for Advanced Scientific Research, Bangalore, India
5   CNRS, Inserm, UMR 7104, Microarray and Sequencing Platform, Institut de Génétique et de Biologie Moléculaire et Cellulaire (IGBMC), Université de Strasbourg, Illkirch, France
6   Laboratory of Biochemistry and Molecular Biology, Hôpital de Hautepierre, University Hospital of Strasbourg, Strasbourg, France
7   Centre de Psychiatrie et Neurosciences, INSERM UMRS894, Université Paris Descartes, Sorbonne Paris Cité, Paris, France
8   Centre de Recherches sur la Cognition Animale, Centre de Biologie Intégrative, CNRS, UPS, Université de Toulouse, Toulouse, France
9   Inserm, CHU-Lille, UMR-S 1172, Alzheimer & Tauopathies, Université de Lille, Lille, France
    *Corresponding author. Tel: +91 80 2208 2840; E-mail: tapas@jncasr.ac.in
    **Corresponding author. Tel: +33 3688 51934; E-mail: laurette@unistra.fr
    †These authors contributed equally to this work

brain tissues from animal models or patients affected by neurodegenerative diseases, with pioneering work coming from studies of Huntington's disease (e.g., Vashishtha *et al*, 2013; Achour *et al*, 2015; Bai *et al*, 2015; Guiretti *et al*, 2016), and only two studies in AD models (Benito *et al*, 2015; Gjoneska *et al*, 2015). Tsai *et al* performed a profiling of transcriptional and epigenomic changes in the hippocampus of a mouse model of AD and showed reduction in H3K27 acetylation (H3K27ac) at downregulated genes which were enriched in synaptic plasticity genes, as well as increased H3K27ac at immune response-enriched upregulated genes (Gjoneska *et al*, 2015). Fischer *et al* tested the effects of the HDAC inhibitor suberoylanilide hydroxamic acid (SAHA) and showed it could reinstate physiological exon usage associated with H4K12 acetylation and plasticity gene expression in aged neurons, but this was not found in AD neurons, where the beneficial effect of SAHA was only partial (Benito *et al*, 2015). Thus, mechanisms underlying epigenetic changes in AD remain cryptic and the mode of action of potential epigenetic drugs and their consequences in AD remain to be established.

Herein, using genome-wide-scale approaches (ChIP-seq/RNA-seq studies), we have investigated the molecular mechanisms associated with plasticity dysfunctions in the hippocampus of a mouse model with AD-like Tau pathology (THY-Tau22). This mouse strain exhibits neurofibrillary tangles (NFTs), neuroinflammation (Laurent *et al*, 2017), spatial memory deficits (Schindowski *et al*, 2006), impaired long-term depression (LTD; Van der Jeugd *et al*, 2011; Ahmed *et al*, 2015), and altered dendritic spine formation in the hippocampus (Burlot *et al*, 2015). In parallel, we have tested the potential therapeutic application of a newly developed CBP/p300 histone acetyltransferases (HAT) activator molecule (CSP-TTK21), as we recently showed that this molecule was able to acetylate nuclear chromatin in the mouse brain (Chatterjee *et al*, 2013). Indeed, as an alternative to HDAC inhibitors, targeting CBP/p300 HATs has been proposed as an innovative therapeutic strategy for disorders affecting memory (Schneider *et al*, 2013; Valor *et al*, 2013). Indeed, CBP loss of function has been reported during neurodegeneration (Rouaux *et al*, 2003, 2004) and CBP/p300 HATs play important roles in neuronal plasticity and cognition, including hippocampal long-term potentiation (LTP) and long-term memory (Alarcon *et al*, 2004; Vecsey *et al*, 2007; Barrett *et al*, 2011; Valor *et al*, 2011; West & Greenberg, 2011).

In this study, we found that treatment of 8-month-old THY-Tau22 mice with CSP-TTK21 restored spatial memory and plasticity functions. At this age, the hippocampal transcriptome induced by spatial learning was severely impaired and CSP-TTK21 treatment re-established the expression of genes involved in neuronal plasticity (immediate early genes, IEGs) and of cognitive enhancers (e.g., Neurotensin, Klotho). Further, we found a specific decrease in H2B (H2Bac)—but not H3K27 (H3K27ac)—acetylation levels in the hippocampus of THY-Tau22 mice, and CBP/p300 activation with CSP-TTK21 significantly restored this signature, notably by a wide re-acetylation of H2B at transcription start sites (TSS) and CBP enhancers. Together, these data indicate that CBP/p300 HAT activation restores an epigenetic landscape that permits, either directly or indirectly, the upregulation of genes involved in neuronal plasticity (IEGs) and memory during learning, thus representing a promising therapeutic option for plasticity and memory enhancement.

# Results

## The CBP pathway is a potential therapeutic target in a mouse model of tauopathy

In the THY-Tau22 mouse model, the tauopathy rapidly progresses once inflammation processes have started, around 7 months of age. Twelve-month-old THY-Tau22 mice exhibit strong tauopathy, with massive accumulation of abnormal Tau conformation and phosphorylation notably in the CA1 region of dorsal hippocampus (Schindowski *et al*, 2006). We found that neurons exhibiting pathological Tau hallmarks (AT100-positive neurons) showed depleted CBP protein levels (Fig 1A). Overall, we measured a global CBP reduction while neuronal marker NeuN levels remained unchanged, but this was associated with an increase in astrogliosis as expected at this age (Fig 1B). Several histone acetylation targets were evaluated by Western blot analyses, and we measured a decrease in H2B tetra-acetylation (H2BK5K12K15K20ac), whereas the global level of the other modifications tested was unchanged (Fig EV1A). In order to test the potential effect of our new drug, we carried out the study at 8 months of age, an earlier symptomatic age where mice already show memory deficits and inflammatory processes while pathology still progresses (Schindowski *et al*, 2006). Transcriptomic analyses performed in the dorsal hippocampus revealed moderate changes (51 dysregulated genes), but confirmed the presence of the inflammatory response signature (Fig 1C). In addition, a series of immediate early genes (IEGs) were downregulated, suggesting reduced basal neuronal activity in the hippocampus of THY-Tau22 vs. WT mice (Fig 1C). Lastly, predicted promoter motifs associated with the 15 downregulated genes were associated with cAMP pathway (ATF2/6 and CREB1; Fig 1C), which is a major regulator of IEGs. Thus, we aimed to test CSP-TTK21 molecule treatment as therapeutic intervention in these mice.

Spatial reference memory tested in the Morris water maze (MWM) is a hippocampus-dependent memory task, in which 8-month-old THY-Tau22 mice fail when the retention test is delayed (Fig 1D). We injected mice three times with CSP-TTK21 (TAU MOL) or the control vehicle (TAU VEH) prior to spatial training (5-day acquisition) and tested mice at a 10-day post-training delay. WT littermates injected with saline vehicle (WT VEH) served as learning control. All mice showed similar acquisition performances in escape latencies (Fig 1D) or distance to the platform (Fig EV1D). Other parameters were also identical (habituation cued trial, swim speed; Fig EV1C and E). However, TAU VEH mice showed a clear retention deficit in the probe test (WT VEH: 23.4 vs. TAU VEH: 15.9 s in the target quadrant), which was prevented when the THY-Tau22 mice received CSP-TTK21 treatment before training (TAU MOL: 23.4 vs. TAU VEH: 15.9 s in the target quadrant; Fig 1D). CSP-TTK21-treated THY-Tau22 mice were also more precise when assessing the latency to first visit to the target quadrant (Fig EV1F) and the search strategy (see Closest to mean and Best tracks Fig EV1H). Only a tendency was measured when assessing platform crossing (Fig EV1G). Thus, CSP-TTK21 treatment fully restored the ability to form long-term memory in the THY-Tau22 mice. Lastly, H2BK5K12K15K20ac and H3K27ac were still more elevated in TAU MOL vs. TAU VEH in the hippocampus of mice euthanized 1 week after the probe test (22 days post-injection, Fig EV1I).

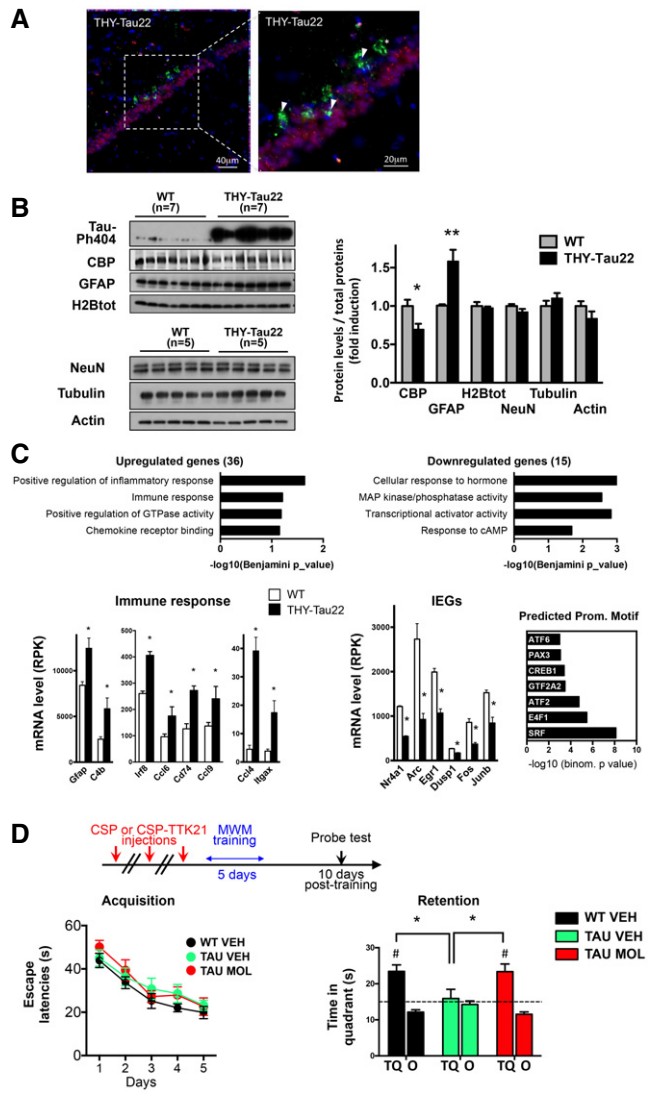

**Figure 1.  Targeting the CBP/p300 pathway restores long-term spatial memory retention in THY-Tau22 mice.**

A    Immunohistochemistry performed on 12-month-old THY-Tau22 mice showing AT100-positive neurons (green), CBP-positive nuclei (red), and DAPI-stained nuclei (blue) in the CA1 region of the dorsal hippocampus. A representative image is shown with a focus. Scale bars: 40 and 20 µm as noted (n = 5 mice). Arrows depict neurons bearing neurofibrillary tangles and do not display CBP immunostaining. The star depicts a ghost tangle (AT100-positive neuron and no nucleus).

B    Western blot analyses performed in the dorsal hippocampus of 12-month-old THY-Tau22 mice compared to age-matched controls. NeuN, actin, tubulin, and total H2B levels are not changed. Phosphorylated Tau on serine 404 (Tau-Ph404) attests to samples from tauopathic mice. Quantification represents the ratio of the protein level detected on the total amount of proteins on the membranes, with WT arbitrarily set at 1 (fold induction). Bar graphs are mean ± SEM. n = 5–7/group as noted, multiple t-tests, and CBP *P = 0.003 and GFAP **P = 0.0001 for THY-Tau vs. WT mice.

C    RNA-sequencing data performed in the hippocampus of 8-month-old mice. Functional analyses (DAVID GOTERM) performed on significantly deregulated genes in THY-Tau22 mice compared to WT mice. Representative genes are presented below. |log2 Fold Change| > 0.2; * indicates FDR < 0.05. IEG, immediate early genes. Predicted promoter motif (right) analyses performed with GREAT. Groups: WT mice (n = 3); THY-Tau22 mice (n = 2). Graphs are mean ± SEM.

D    Long-term spatial memory testing: Mice were injected three times (1 per week) with vehicle (WT mice saline, WT VEH, n = 17), vehicle (THY-Tau22 mice CSP, 500 µg/mouse, TAU VEH, n = 10), or molecule (THY-Tau22 mice CSP-TTK21, 500 µg/mice TAU MOL, n = 13) before training of spatial memory in the Morris water maze (MWM); retention (Probe test) was tested 10 days after the last training session. Acquisition (escape latencies, seconds) and retention performances (time in target quadrant, seconds) are shown for the three groups of mice. All groups of mice displayed significant acquisition of the platform location [Day effect, F(4,148) = 26.45, P < 0.001; Group and Group × Day effects, ns], but only TAU VEH exhibited impaired retention. CSP-TTK21 treatment fully restored the ability of THY-Tau22 mice to remember the platform location. Bar graphs are mean ± SEM. Student's t-test to a constant value, # when compared to random (dotted line, 15 s) WT VEH, t(16) = 4.6323, #P = 0.0002; TAU VEH, t(9) = 0.3606, P = 0.7267; TAU MOL, t(12) = 3.945, #P = 0.0019. One-way ANOVA; F(2,37) = 3.55; P = 0.03; * in the different comparisons: TAU MOL vs. TAU VEH, *P = 0.0166; WT VEH vs. TAU VEH, *P = 0.040; TAU MOL vs. WT VEH, non-significant P = 0.437. TQ, target quadrant; O, other corresponds to the mean of the three other quadrants.

## Activation of the CBP/p300 HAT pathway restores synaptic plasticity in the hippocampus of THY-Tau22 mice

We next determined whether CSP-TTK21 treatment could affect structural plasticity such as dendritic spine formation. Spines can be identified based on their morphological appearance. Spines with no head such as filopodia are considered immature. Stubby spines show a protrusion but no head or neck and are less mature than headed spines (thins and mushrooms) according to Harris *et al* (1992). Mushroom spines are thought to be stabilized by learning processes to form new synapses (Restivo *et al*, 2009; Caroni *et al*, 2014). We first tested the effect of CSP-TTK21 injections in the absence of learning, in the CA1 regions of THY-Tau22 mice, that shows diminished levels of total spines (Burlot *et al*, 2015): A single CSP-TTK21 injection was able to increase the density of total spines, with a significant impact on the stubby and filopodia sub-types (immature spines, Fig 2A). We next tested the effect of CSP-TTK21 injections in response to spatial training. Learning requires the stabilization, strengthening, and elimination of emerging synapses in a

time-specific manner, and synaptic rearrangements necessary for the long-term consolidation of a memory task require 1–4 days (Caroni *et al*, 2014). Therefore, mice were treated with CSP-TTK21, trained for 4 days, and mushroom-shaped (i.e., mature) spines were counted 4 days later. An increase in mushroom spines was observed in the WT mice, but not in THY-Tau22 mice. CSP-TTK21 fully restored the learning-induced spine production and maturation in CA1 pyramidal neurons in tauopathic mice (Fig 2B). Lastly, we tested the effect of CSP-TTK21 on long-term depression (LTD), which is altered in THY-Tau22 mice (Van der Jeugd *et al*, 2011; Ahmed *et al*, 2015). CSP-TTK21 treatment rescued LTD maintenance in hippocampal slices (Fig 2C).

## Transcriptomic effects of CSP-TTK21 in learning THY-Tau22 mice

We further checked the effect of CSP-TTK21 on gene expression. As RNA-seq performed on tauopathic mice in basal conditions showed a limited number of differentially expressed genes (Fig 1C), we generated RNA-seq data in CSP and CSP-TTK21-treated THY-Tau22 mice

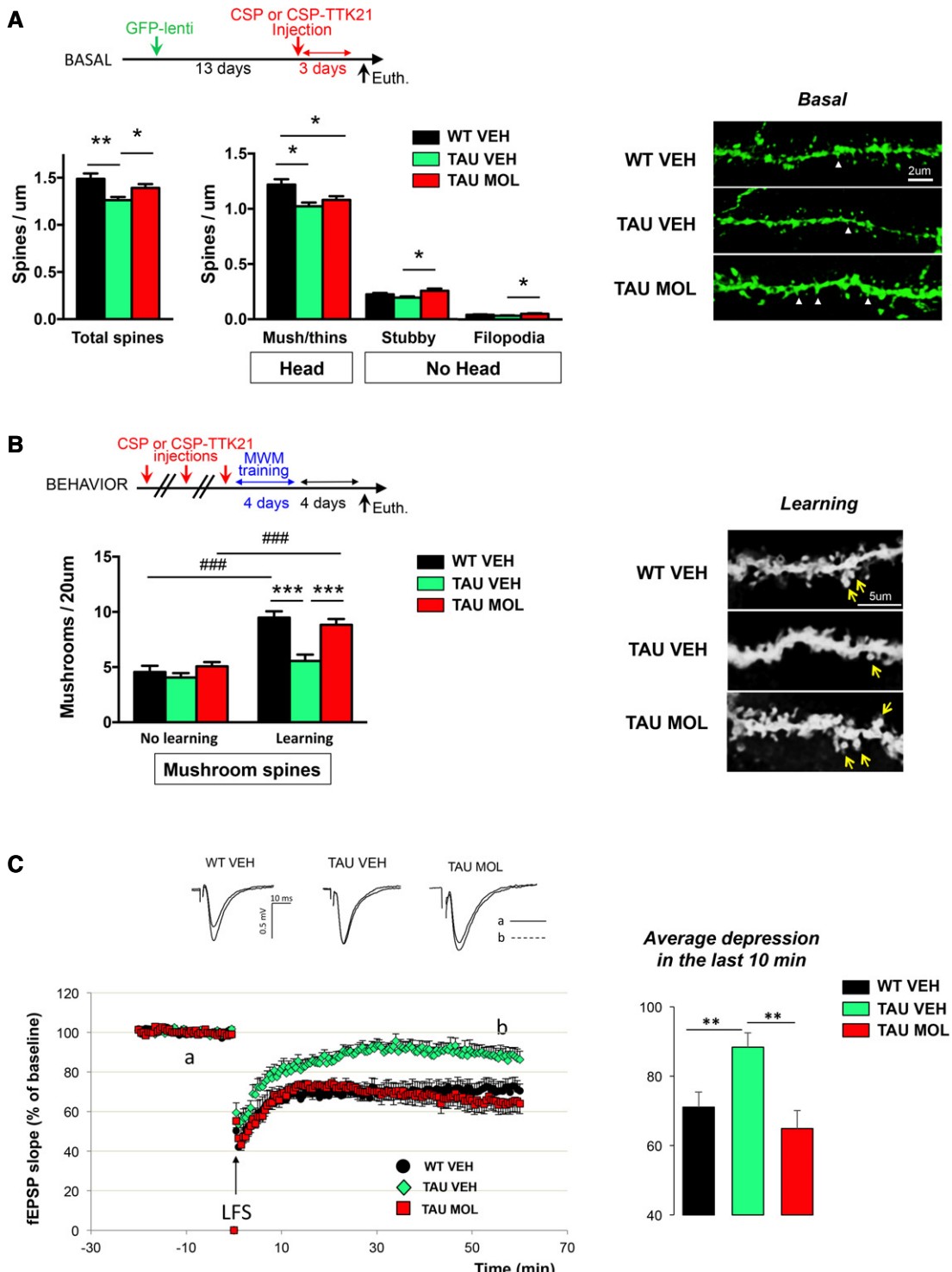

Figure 2.

(respectively, TAU VEH and TAU MOL) subjected to spatial learning; WT mice being the controls (Fig 3A). In response to learning, we found that 2,756 genes were differentially regulated in TAU VEH vs. WT VEH mice (Fig 3B). Expression level of these genes assessed in the different experimental groups showed that the molecule treatment of THY-Tau22 mice (TAU MOL) tended to normalize gene expression toward the WT WEH group (Fig 3C). In fact, only 1,716 genes were still deregulated in TAU MOL vs. WT VEH over the 2,756 ones in TAU VEH vs. WT VEH (Fig 3D), which corresponds to almost 50% rescue of the transcriptome of TAU mice during learning before and after treatment (Fig 3E). Interestingly, 180 genes were directly modulated by the CSP-TTK21 molecule in TAU mice (Fig 3F). These results

**Figure 2.  HAT activation with CSP-TTK21 treatment restores plasticity in the hippocampus of 8-month-old tauopathic mice.**

A  The timeline of GFP-lentivirus and CSP or CSP-TTK21 (one injection, 500 μg/mouse) injection is shown. (Left) The total number of spines was significantly decreased in TAU VEH compared to WT VEH mice. TAU MOL hippocampi showed a significant increase in the total number of spines compared to TAU VEH (one-way ANOVA $P = 0.0011$, *post hoc* Holm–Sidak multiple-comparisons test $F(2,243) = 7.03$; WT VEH vs. TAU VEH, **$P < 0.0009$; TAU MOL vs. TAU VEH, *$P = 0.0455$). (Middle) Based on spine type, the number of head spines (mushrooms, thins) was significantly lower in both TAU VEH and TAU MOL than in WT VEH controls (one-way ANOVA $P = 0.0018$, *post hoc* Holm–Sidak multiple-comparisons test $F(2,243) = 6.467$; WT VEH vs. TAU VEH, *$P = 0.0013$; WT VEH vs. TAU MOL, *$P = 0.0272$. Stubby spine density was significantly increased in TAU MOL compared to TAU VEH mice (one-way ANOVA $F(2,243) = 4.311$, $P = 0.0145$, *post hoc* Holm–Sidak multiple-comparisons test: TAU MOL vs. TAU VEH, *$P = 0.0109$), as the number of filopodia (one-way ANOVA $F(2,243) = 3.845$, $P = 0.0227$, *post hoc* Holm–Sidak multiple-comparisons test TAU MOL vs. TAU VEH, *$P = 0.0179$). (Right) Typical images are presented showing a dendrite fragment for each condition. White arrowhead depicts stubby spines. Scale bar, 2 μm. Number of dendritic segments: WT VEH, $n = 67$; TAU VEH, $n = 93$, TAU MOL, $n = 87$; number of neurons: WT VEH, $n = 20$; TAU VEH, $n = 28$, TAU MOL, $n = 16$; number of mice: WT VEH, $n = 2$; TAU VEH, $n = 3$, TAU MOL, $n = 3$.

B  CSP-TTK21 injection into THY-Tau22 mice rescues mature dendritic spines formation in response to learning. (Top left) The timeline of injections is shown: Mice were injected three times (1 per week) with vehicle (WT mice, WT VEH, NaCl 0.9%), vehicle [THY-Tau22 mice (TAU VEH), CSP 500 μg/mouse], or molecule [THY-Tau22 mice (TAU MOL), 500 μg/mice] and either trained over a 4-day acquisition period in the MWM ("learning" group) or left in their home cage ("basal" group). Mushroom-shaped spines were counted in dorsal CA1, 4 days post-training. (Bottom left) The number of mature spines was significantly increased by learning in WT VEH and TAU MOL mice. Two-way ANOVA; learning effect, $F(1,139) = 54.18$; $P < 0.0001$; ### *post hoc* Holm–Sidak multiple-comparisons test: learning vs. basal in WT VEH ($P = 0.0001$) and in TAU MOL mice ($P = 0.0001$). After learning, WT VEH and TAU MOL mice displayed significantly higher number of mature spines than TAU VEH mice (Genotype X Treatment effect, $F(2,139) = 9.704$; $P = 0.0001$; *** *post hoc* Holm–Sidak multiple-comparisons test: TAU MOL vs. TAU VEH ($P = 0.0001$), WT VEH vs. TAU VEH ($P = 0.0001$). (Right) Typical examples of a dendritic fragment bearing mushroom spines (arrows) are shown for each sub-group in response to learning. Number of dendritic segments: *Learning*: WT VEH, $n = 27$; TAU VEH, $n = 27$, TAU MOL, $n = 30$; WT VEH_HC, $n = 27$; number of mice: WT VEH, $n = 3$; TAU VEH, $n = 3$, TAU MOL, $n = 3$. *Basal*: WT VEH, $n = 27$; TAU VEH, $n = 7$, TAU MOL, $n = 15$; number of mice: WT VEH, $n = 3$; TAU VEH, $n = 3$, TAU MOL, $n = 2$.

C  Mice were injected three times (1 per week) with saline (WT VEH), CSP (vehicle, VEH), or CSP-TTK21 (molecule, MOL) (500 μg/mice; THY-Tau22 mice (TAU) before euthanasia. Long-term depression measurements were performed on hippocampal slices. (Top left) Examples of analog traces recorded 10 min before (*a*) and 55 min after LTD induction (*b*; dotted line) in the three groups of mice. (Bottom left) Time course of LTD; LTD is expressed as a percent change in fEPSP (field excitatory postsynaptic potentials) slope over time. After the 20-min baseline recording, a low-frequency stimulation (LFS, 2 Hz for 10 min) was applied (arrow). Recording was stopped during the 10-min conditioning stimulation and resumed after completion of LFS. LFS induced a strong depression of the fEPSP slope, which recovered partially to reach a stable level of depression about 20 min after stimulation. (Right) Average depression measured in the last 10 min of LTD. LTD was significantly different in TAU VEH ($88.4 \pm 4.1\%$ of the baseline, $n = 10$) compared to controls (WT VEH, $71.1 \pm 4.4\%$, $n = 9$; $F(1,17) = 8.8$, **$P = 0.008$). CSP-TTK21 treatment restored LTD to control levels ($64.9 \pm 5.2\%$, $n = 10$; WT VEH vs. TAU MOL: $F(1,17) = 0.83$, $P = 0.37$, ns; TAU VEH vs. TAU MOL: $F(1,18) = 13.2$. **$P = 0.0019$). Multivariate analyses of variance followed by *post hoc* test (Statview software).

Data information: Graphs are mean ± SEM.

suggest that CSP-TTK21 induced direct and indirect mechanisms that act on the transcriptome. Lastly, learning itself induced the differential regulation of 1,663 genes in WT mice (Fig 3G). When up- and downregulated genes were analyzed separately in TAU vs. WT mice (1,966 genes DOWN and 790 genes UP), fold changes observed tended toward normalization after treatment (all the genes, Fig 3H; see the 300 most deregulated genes, Fig EV2A). Functional Biological Process annotation showed that downregulated genes in tauopathy were associated with ion transport, transcription, and synaptic plasticity, whereas upregulated ones were associated with protein catabolism (e.g. transport, ubiquitination). The treatment dramatically lowered the significance of each of these pathways (Fig 3H). However,

CSP-TTK21 had no effect on inflammatory processes (Fig EV2B and C) and did not affect the transgene expression (Fig EV2D).

We then evaluated direct effects of the molecule and analyzed the 98 significantly upregulated genes separately (TAU MOL vs. TAU VEH; Fig 4). Functional enrichment analysis revealed that they were associated with the cellular response to cAMP, which is in agreement with the expected effect of a CBP/p300 activator (Fig 4A). They also associated with ion-related processes (binding, transport, and channels), suggesting that CSP-TTK21 rescued part of the ion dyshomeostasis in the hippocampus of THY-Tau22 mice. The average expression of these 98 upregulated genes in the TAU MOL vs. TAU VEH comparison was then represented as a heatmap in all

**Figure 3.  CSP-TTK21 injections of Tauopathic mice prior to learning and memory re-established part of the deregulated transcriptome.**

A  Timeline of the experiment and experimental groups: Eight-month-old mice were injected three times (1 per week) with vehicle (WT mice, WT VEH, saline), vehicle [THY-Tau22 mice (TAU VEH), CSP 500 μg/mouse], or molecule [THY-Tau22 mice (TAU MOL), CSP-TTK21 500 μg/mice]. One sub-group of WT mice was left in their home cage (WT VEH_HC), and the other groups of mice (WT and THY-Tau22) were subjected to a 3-day spatial training (Learning). RNA extracts were isolated from the dorsal hippocampus, 1 h after the last training ($n = 5$/group).

B  Volcano plot showing that 2,756 genes are differentially regulated between THY-Tau22 and WT mice during learning. The $\log_2$(Fold-Change) was estimated by DESeq2. FDR < 0.05 and |log2 Fold Change| > 0.2. Red dots correspond to genes with adjusted *P*-value < 0.05. Numbers in the corners represent the number of downregulated (left) and upregulated (right) genes.

C  Heatmap representing expression of *z*-score of the 2,756 deregulated genes in all experimental conditions. Color coding was performed according to the *z*-score of the normalized reads counts divided by gene length. Clustering was performed using the unweighted pair group method with arithmetic mean method and Pearson's distance. Groups: WT VEH_HC ($n = 3$), WT VEH ($n = 2$); TAU VEH ($n = 2$); TAU MOL ($n = 2$).

D  Volcano plot showing that 1,756 genes are differentially regulated between treated THY-Tau22 and WT mice during learning. Same parameters as in (B).

E  Venn diagram showing the rescue of the transcriptome in learning mice, by comparing differentially regulated genes before (TAU VEH vs. WT VEH) and after (TAU MOL vs. WT VEH) the treatment for all genes with an adjusted *P*-value < 0.05.

F, G  Volcano plots showing differentially regulated genes by the molecule in THY-Tau22 mice (F) and by learning in WT mice (G). Same parameters as in (B).

H  Fold changes for every significant pathology-regulated gene (TAU VEH vs. WT VEH) condition are plotted for downregulations (top) and upregulations (bottom) separately (Adjusted *P*-value < 0.05). Functional annotation performed with DAVID for GOTERM_Biological Process and their significance are shown on the right before (green) and after (red) treatment. Significantly regulated pathways with −log10(Benjamini P_value) < 0.05 are given.

experimental groups (Fig 4B), showing that TAU MOL samples were closer to WT VEH ones in learning mice. The expression of these genes was very low in the TAU VEH mice and clustered with non-behaving WT mice (WT VEH_HC). Not only the large majority of the 98-induced genes overlapped with those deregulated in the pathology

(81/98; Fig 4C), but a significant proportion of them was also upregulated in learning conditions (34/98; Fig 4C), providing evidence that CSP-TTK21 significantly rescued the expression of genes that are specifically involved in learning and memory. In line, CSP-TTK21 induced the expression of a series of IEGs (e.g., *Arc*, *c-fos*, and *egr1*)

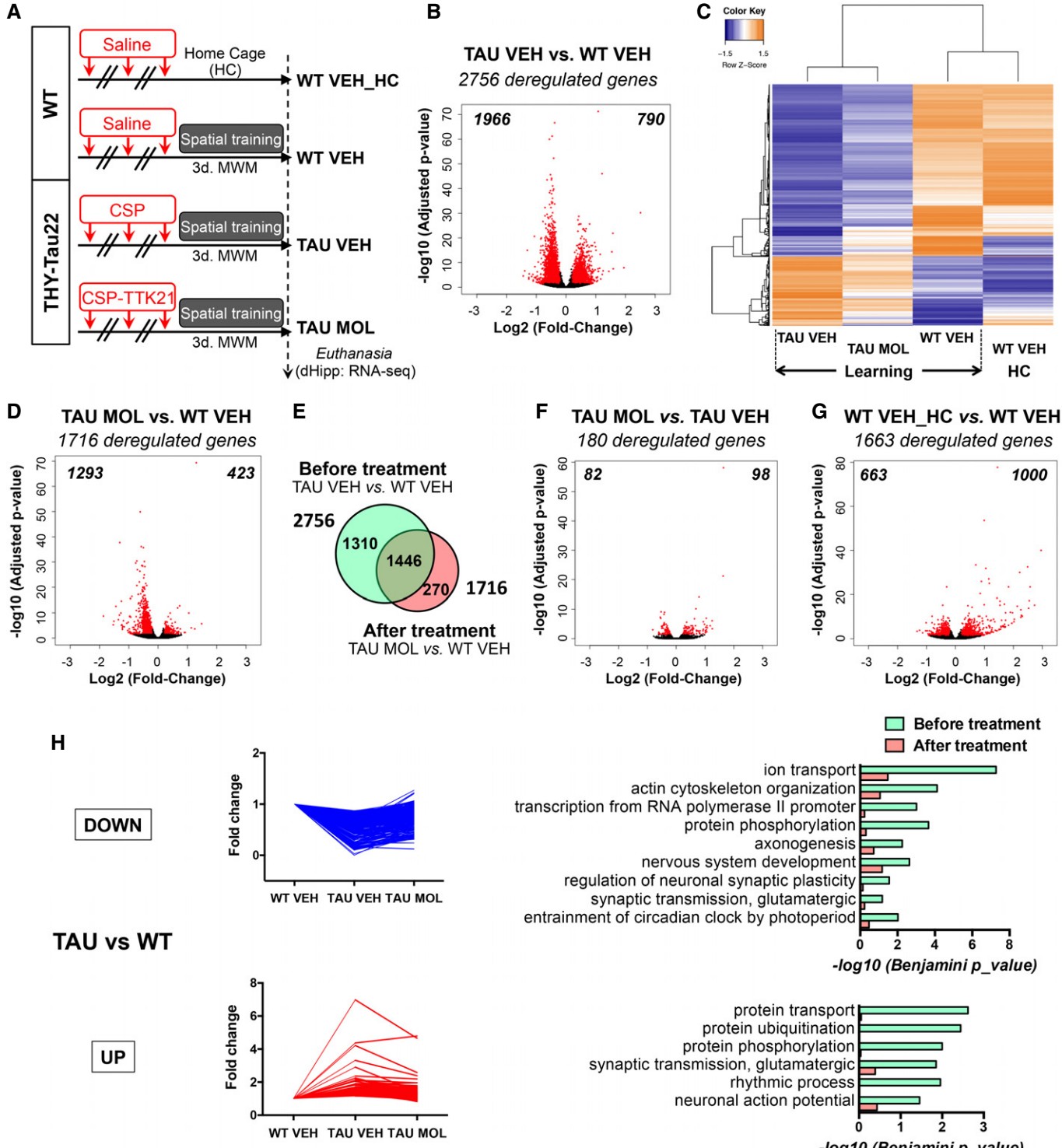

**Figure 3.**

in both basal and learning conditions (Fig 4D; Appendix Fig S1D and E), suggesting that the molecule improved neuronal activity. Last, decreased transcription of target genes, such as *Klotho (Kl)* and *Neurotensin (Nts)*, in THY-Tau22 vs. WT mice was also counterbalanced by CSP-TTK21 treatment in THY-Tau22 mice, up to the protein level (Fig 4E–G). Since CSP-TTK21 did not rescue inflammatory markers (Fig EV2A), our results further indicate that CSP-TTK21 rather modulated the expression of genes involved in plasticity and learning and memory processes, than it acted on the pathology *per se*.

We also found that CSP-TTK21 reduced the expression of a significant number of genes associating with neuronal phenotype in Cellular Component annotations (Fig 5A–C). Approximately half of them (40/82; Fig 5D) were found upregulated in THY-Tau22 vs. WT mice, and 10% (9/82) significantly downregulated by learning (Fig 5D). Interestingly, this functional signature also characterized the 663 downregulated genes in WT mice in response to learning (Fig 5E), suggesting that learning induces downregulation of neuronal genes in the dorsal hippocampus. This contrasts with the 1,000 upregulated genes by learning, which were associated with transcriptional and translational (ribosome and reticulum) processes (Fig 5F).

Altogether, our transcriptomic data support a rescued pathological phenotype likely through activation of immediate early genes restoring neuronal activity and induction of proteins supporting cognitive enhancement, leading to a partial restoration of the transcriptome regulated during learning.

## H2B acetylation epigenetic signature is disrupted in THY-Tau22 mice

As CSP-TTK21 activates CBP/p300 HAT function, we next investigated whether it could modulate the epigenome in THY-Tau22 mice. Among the different histones, H2B is a CBP-target in the hippocampus (Alarcon *et al*, 2004; Barrett *et al*, 2011; Valor *et al*, 2011). In addition, we previously showed that H2B acetylation (H2Bac) levels were increased in the hippocampus of memory-trained rats, particularly at the proximal promoter peaks of IEGs (including *Fos* and *Egr-1*; Bousiges *et al*, 2010, 2013). Further, CBP/p300 HAT activation with CSP-TTK21 molecule increased H2Bac in the hippocampus of WT mice and induced persistence of spatial memory (Chatterjee *et al*, 2013). Together, these data suggest that H2Bac levels might represent a key regulator of hippocampal gene expression both in resting and in learning conditions. We thus analyzed the distribution of this histone mark by ChIP-sequencing in three groups of resting mice: WT VEH, TAU VEH, and TAU MOL. Duplicates were performed, and we also immunoprecipitated H3K27ac (Fig EV3A and B), another known target of CBP (Jin *et al*, 2011; Tie *et al*, 2014). Compared to the whole genomic distribution, H2Bac peaks were mainly enriched in gene profiles (introns, 42 to 29%) and promoter regions (6 to 2%) to the cost of intergenic regions (29 to 46%; Fig 6A). These profiles were similar to that of H3K27ac except that H2Bac was less enriched at promoter regions (6% vs. 12%). Interestingly, H2Bac peaks were always detected when there was H3K27ac and 43% of H3K27ac- and H2Bac-covered nucleotides colocalized. H2Bac was strongly associated with the TSS of highly expressed genes (Q4; Fig 6B) and gene bodies, whose main annotations were related to neuronal signaling (Fig 6C). In contrast, H2Bac was poorly found on genes related to the olfactory system that are repressed in the hippocampus (Fig EV3C). We then compared H2Bac enrichment in the different experimental conditions: WT VEH, TAU VEH, and TAU MOL. When THY-Tau22 mice were compared to WT mice, we found mainly decreased H2Bac levels, at 1,624 peaks (associated with 1,338 genes), whereas only a few peaks (14) showed H2Bac enrichment (Fig 6D). After CSP-TTK21 injection, H2Bac was increased at 2,617 peaks (associated with 1,984 genes) in THY-Tau22 mice, whereas it was decreased at only four peaks, indicating a substantial activation of histone acetyltransferase activity in response to CSP-TTK21 treatment (Fig 6D). Importantly, increased acetylation levels at the 1,624 decreased peaks

---

**Figure 4.   CSP-TTK21 injections induced the expression of 98 genes in tauopathic mice during learning.**

A    Functional annotation charts (Benjamini, $P < 0.05$) using GREAT performed on the 98 significantly overexpressed genes by CSP-TTK21 treatment in tauopathic mice (TAU MOL vs. TAU VEH). Significance is indicated as $-\log10$(Benjamini P_value).

B    Heatmap representing expression *z*-score of the 98 significantly overexpressed genes in the TAU MOL vs. TAU VEH comparison, for all experimental conditions. Color coding was performed according to the *z*-score of the normalized reads counts divided by gene length. Clustering was performed using the unweighted pair group method with arithmetic mean method and Pearson's distance.

C    Venn diagram showing that most of the genes upregulated by CSP-TTK21 treatment were also downregulated in the pathology (81/98, $P = 7.1 \times 10^{-88}$, $\chi^2$ test), as well as a significant overlap ($P = 2 \times 10^{-29}$, $\chi^2$ test) genes upregulated by learning in WT.

D    RNA-seq data showing expression of several immediate early genes (IEGs): Nr4a1, Arc, Egr1, Dusp1, Fos, and Junb. Most of the downregulated IEGs in tauopathic compared to WT mice present a significant induction in THY-Tau22 mice after CSP-TTK21 injection. Adjusted *P*-values corresponding to $P < 0.05$ are indicated by * for WT VEH vs. WT VEH_HC, $ for TAU VEH vs. WT VEH, and # for TAU MOL vs. TAU VEH. See Materials and Methods for statistical analyses of RNA-seq data ($n = 2$–3/group).

E    RNA-seq data showing that CSP-TTK21 treatment of tauopathic mice restored the expression of plasticity/memory-relevant target genes: *Klotho (Kl)* and *Neurotensin (Nts)*. Adjusted *P*-values corresponding to $P < 0.05$ are indicated by * for WT VEH vs. WT VEH_HC, $ for TAU VEH vs. WT VEH and # for TAU MOL vs. TAU VEH. See Materials and Methods for statistical analyses of RNA-seq data ($n = 2$–3/group).

F    RT–qPCR validations performed in a different cohort of mice from the RNA-seq study ($n = 4$–5/group). One-way ANOVA with uncorrected Fisher's test. *Klotho, Kl*: $F_{(3,16)} = 2.949$, $P = 0.0643$; * learning $P = 0.0145$ for WT VEH vs. WT VEH_HC, $ pathology $P = 0.0395$ for WT VEH vs. TAU VEH. *Nts*: $F_{(3,14)} = 4.290$, $P = 0.0241$; $ pathology $P = 0.0036$ for WT VEH vs. TAU VEH, # molecule $P = 0.0081$ for TAU MOL vs. TAU VEH.

G    Western blot for Klotho and Neurotensin expression in the different experimental conditions. Immunoblot results are shown ($n = 5$/group). Bar graphs represent the quantification of the protein levels as percentage of the control group WT VEH_HC, arbitrarily set at 100%. Each detected protein was normalized to the corresponding amount of total proteins in the gels or the nitrocellulose membrane. Klotho and Neurotensin were further normalized to the level of actin. One-way ANOVA with uncorrected Fisher's test. Klotho: $F_{(3,16)} = 5.192$, $P = 0.0107$; * learning $P = 0.0117$ for WT VEH vs. WT VEH_HC, $ pathology $P = 0.0027$ for WT VEH vs. TAU VEH, # molecule $P = 0.0282$ for TAU MOL vs. TAU VEH. Nts: $F_{(3,16)} = 3.748$, $P = 0.0326$; $ pathology $P = 0.0200$ for WT VEH vs. TAU VEH, # molecule $P = 0.0079$ for TAU MOL vs. TAU VEH.

Data information: Graphs are mean ± SEM.
Source data are available online for this figure.

were found after CSP-TTK21 treatment (Figs 6E and F, and EV3D). By contrast, H3K27ac levels did not show any modulation at these sites in the different experimental conditions (Fig 6E), indicating a specific signature on H2Bac in THY-Tau22 mice. Functional enrichment analyses revealed that peaks showing decreased H2Bac were associated with genes involved in neuronal signaling pathways (cAMP, cGMP, calcium, MAPK), LTP/LTD, and in synapses, and so were the peaks showing increased H2Bac upon CSP-TTK21 treatment (Fig 6G). These data suggest that CSP-TTK21 re-acetylated a number of gene loci involved in neuronal functions, including

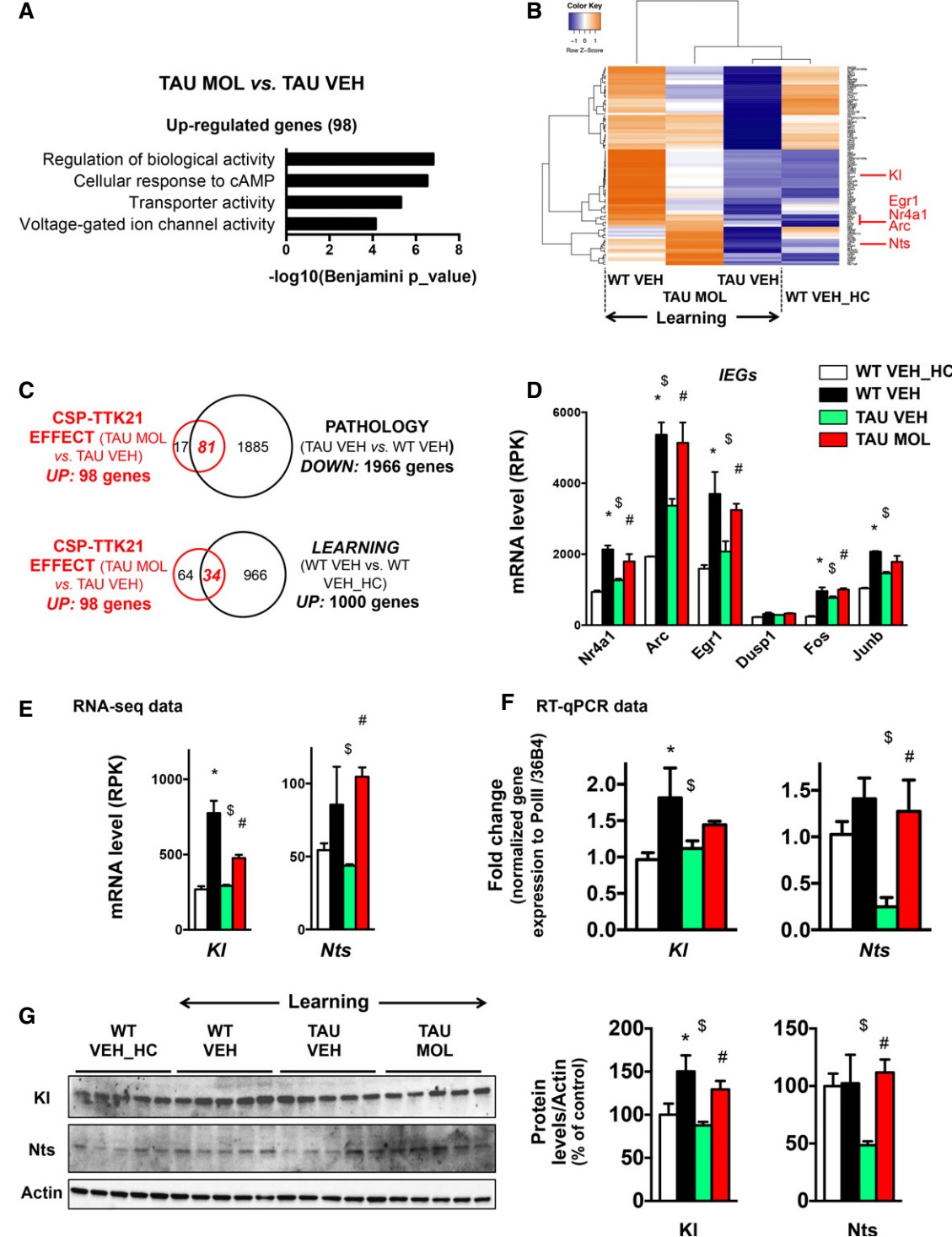

**Figure 4.**

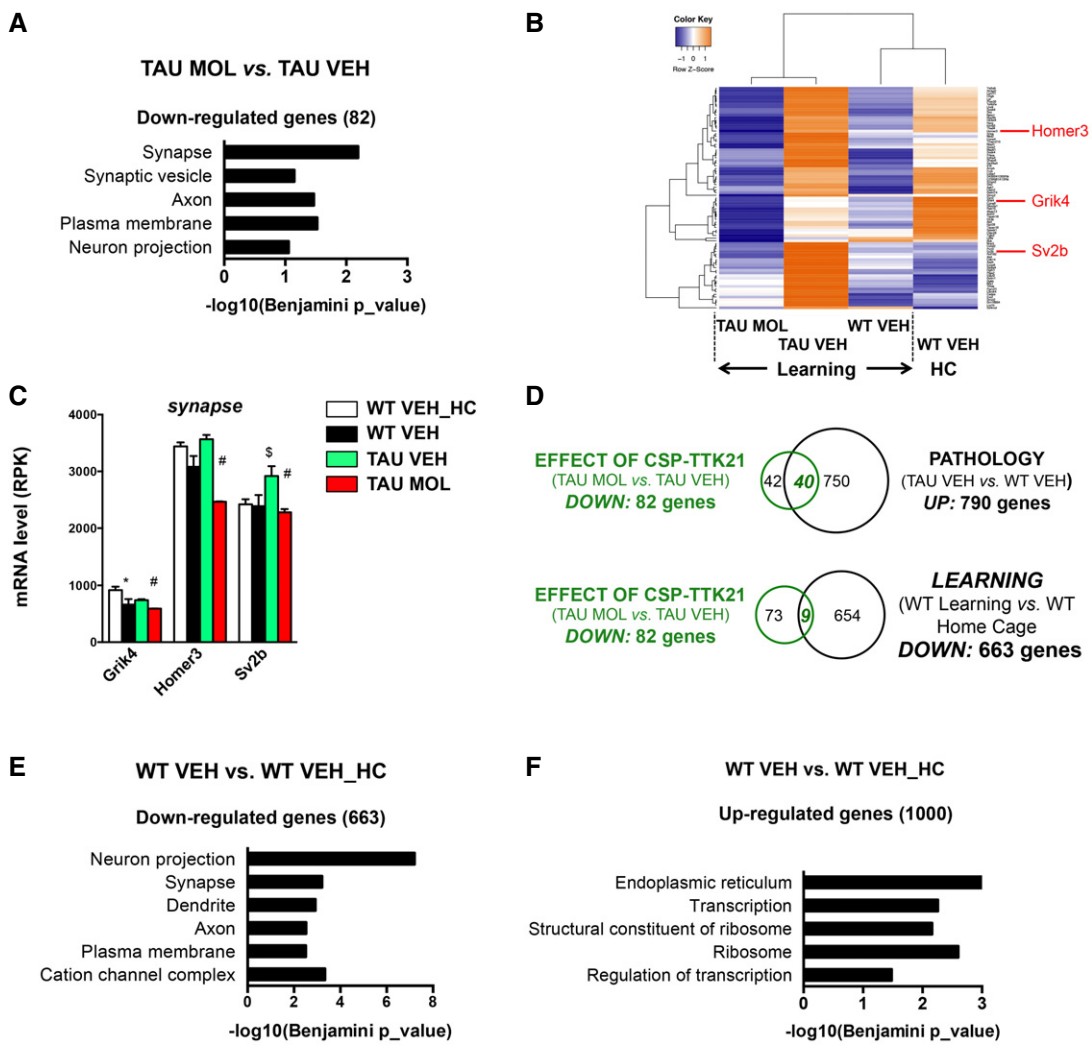

**Figure 5. CSP-TTK21 injections repress the expression of 82 genes in tauopathic mice during learning.**

A    Functional annotation charts (Benjamini, $P < 0.05$) using DAVID (GOTERM_Cellular Component) performed on the 82 significantly downregulated genes by CSP-TTK21 treatment in tauopathic mice (TAU MOL vs. TAU VEH), showing a significant association with neuronal terms. Significance is indicated as −log10(Benjamini $P$_value).

B    Heatmap representing expression *z*-score of the 82 significantly downregulated genes in the TAU MOL vs. TAU VEH comparison, for all experimental conditions. Color coding was performed according to the *z*-score of the normalized reads counts divided by gene length. Clustering was performed using the unweighted pair group method with arithmetic mean method and Pearson's distance.

C    RNA-seq data showing expression of several genes belonging to "synapse" annotation (DAVID): *Grik4*, *Homer3*, and *Sv2b*, which are significantly downregulated by CSPTTK21 treatment in THY-Tau22 mice. Adjusted *P*-values corresponding to $P < 0.05$ are indicated by * for WT VEH vs. WT VEH_HC, $ for TAU VEH vs. WT VEH, and # for TAU MOL vs. TAU VEH. See Materials and Methods for statistical analyses of RNA-seq data (*n* = 2–3/group). Graphs are mean ± SEM.

D    Venn diagram showing that a significant number of genes downregulated by CSP-TTK21 treatment overlapped with genes upregulated in the pathology ($P = 2.1 \times 10^{-72}$, $\chi^2$ test), as well as with some genes downregulated by learning in WT ($P = 2.3 \times 10^{-3}$, $\chi^2$ test).

E, F    Functional annotation charts using DAVID (GOTERM_Cellular Component) performed on the differentially regulated genes in learning in the WT mice condition (WT VEH vs. WT VEH_HC). The downregulated genes by learning (E) are associated with neuronal terms, and upregulated ones (F), with transcriptional and translational functions. Significance is indicated as −log10(Benjamini $P$_value).

neuronal plasticity. Overall, when differentially acetylated peaks were compared before and after CSP-TTK21 treatment, we found that almost 95% of peaks were rescued (Fig 6H), but quite specifically as only 148 peaks remained decreased and 324 new peaks showed increased acetylation (Fig EV3E).

We next checked the specificity of our pharmacological approach using CBP ChIP-seq data previously generated in cortical cultures

(Kim *et al*, 2010) to determine H2Bac enrichment at CBP enhancers in the dorsal hippocampus of WT mice (Fig 7A). We found that H2Bac was globally enriched at CBP enhancers (Fig 7A; Lopez-Atalaya *et al*, 2013), as well as H3K27ac, as reported in other tissues (Kim *et al*, 2010; Jin *et al*, 2011; Malik *et al*, 2014; Tie *et al*, 2014). Importantly, H2Bac was differentially enriched at CBP enhancers (i.e., decreased H2Bac levels in TAU VEH vs. WT VEH

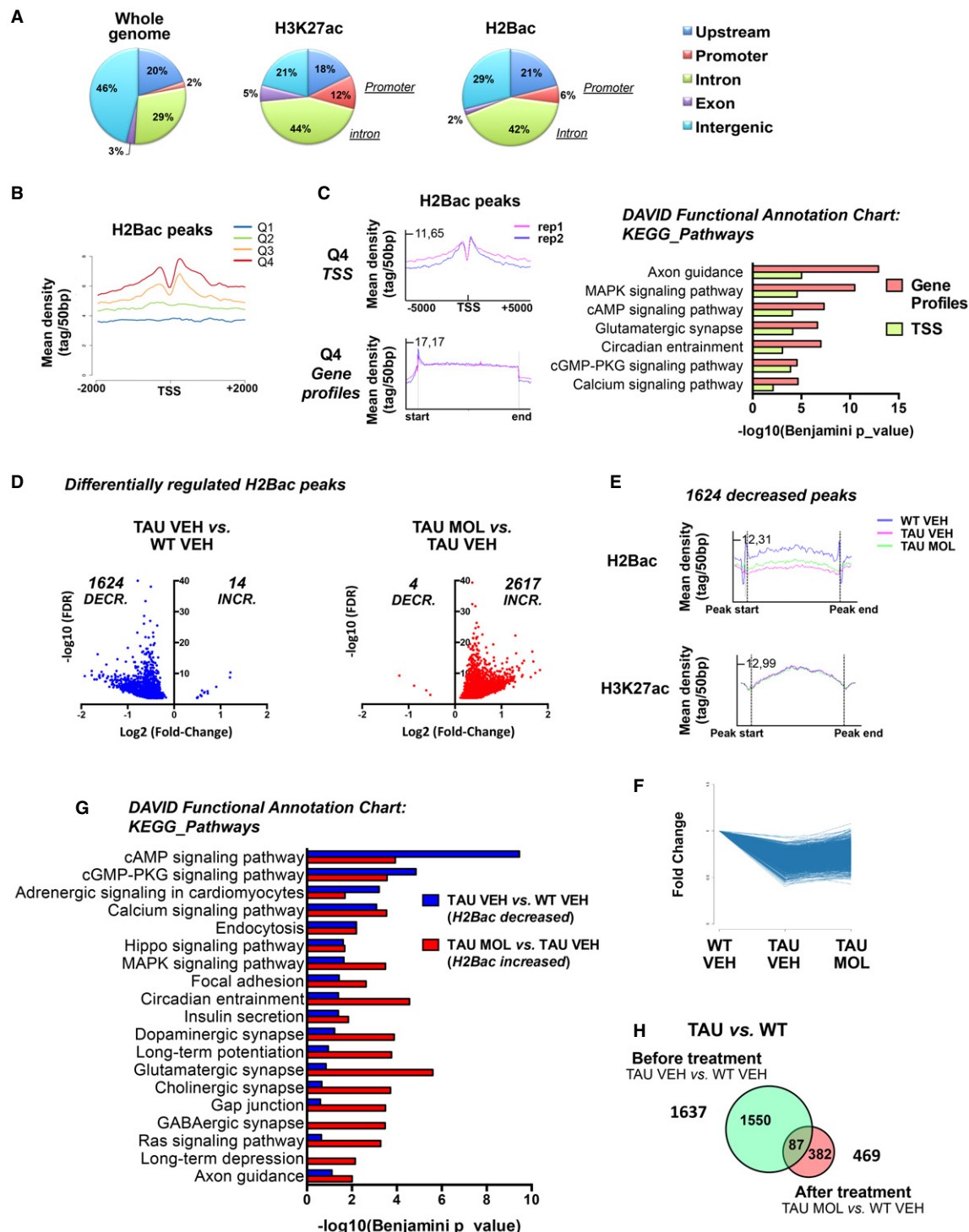

Figure 6.

mice and enriched H2Bac levels in TAU MOL vs. TAU VEH), suggesting that the CSP-TTK21 molecule likely activated the HAT function of CBP *in vivo* to re-acetylate H2B at CBP enhancers. Interestingly, CBP enhancers could be separated into two different classes according to the H2Bac enrichment: highly (17,817) and poorly (23,331) enriched peaks (Fig 7B). Only highly enriched peaks showed a differential regulation of H2Bac, and they were

associated with genes involved mainly in GO terms related to neuronal signaling, synapse, and glutamate activity (Fig 7C). Of note, H3K27ac peaks did not show any modulation on CBP enhancers (Fig 7A). We then found that H2Bac was also differentially enriched on TSS (Fig 7D), with a more pronounced effect on highly expressed genes (fourth quartile; Fig EV4A), and this was not the case for H3K27ac. Integration with transcriptomic studies showed

**Figure 6.   H2Bac levels are severely decreased in tauopathic mice and increased by CSP-TTK21 treatment.**

A   Pie charts showing the genomic distribution of H2Bac and H3K27ac relative to whole genome, obtained from ChIP-sequencing experiments performed in WT dorsal hippocampus. The percentage of peaks in promoters, upstream regulatory regions (upstream; −1 to −20 kb relative to TSS), introns, exons, and intergenic regions is shown.

B   Mean H2Bac profiles at TSS relative to gene expression obtained from RNA-seq data. Expressed genes were separated into four groups (Q1 to Q4, 6,483 genes/group), Q1 representing the less expressed genes (0–25%) and Q4, the most expressed genes (75–100%). Profiles were established with SeqMiner centered on TSS ±2 Kb.

C   Mean H2Bac profiles established with SeqMiner for all genes at TSS (top) and gene bodies (bottom) for the fourth quartile (most expressed genes) in the two replicates (rep1 and rep2). Significance (Benjamini, $P < 0.05$) of the annotations associated with both TSS (green) or gene profiles (red) are shown (DAVID, KEGG pathways).

D   Differentially regulated H2Bac peaks were analyzed with SICER, and significant fold changes (FC > 1; FDR < 0.001) between comparisons are shown for the pathology (TAU VEH vs. WT VEH, blue) and the effect of the molecule (TAU MOL vs. TAU VEH, red). Fold change is shown as log2 (Fold Change) (x-axis) and significance, as −log10(FDR) (y-axis).

E   Mean H2Bac and H3K27ac profiles established with SeqMiner for the 1,624 decreased peaks obtained in the comparison TAU VEH vs. WT VEH. H2Bac but not H3K27ac peaks show differential regulation.

F   Time series-like plot. For every significant H2Bac decreased regions (1,624) in WT VEH vs TAU VEH, normalized read counts were computed for each of the different conditions (WT VEH, TAU VEH, TAU MOL). Fold changes are all referring to WT VEH. Y-axis: Fold change. X-axis: Conditions.

G   Functional annotation charts using DAVID performed on the differentially regulated H2Bac genes corresponding to the peaks either decreased in TAU VEH vs. WT VEH (blue, 1,624) or increased in TAU MOL vs. TAU VEH (red, 2,617). Significance is indicated as −log10(Benjamini P_value).

H   Venn diagram showing the rescue of H2Bac levels by comparing differentially regulated peaks before (TAU VEH vs. WT VEH) and after (TAU MOL vs. TAU VEH) the treatment for all peaks with an FDR < 0.001. About 95% of the peaks were not deregulated after CSP-TTK21 treatment.

that the TSS of differentially regulated genes, be they decreased or increased by learning, also showed a differential H2Bac enrichment (i.e., less H2Bac in THY-Tau22 vs. WT mice and more H2Bac after CSP-TTK21 treatment) in their basal state (Fig 7E). This finding suggests that a H2Bac-enriched TSS acetylation landscape is important for further learning-induced gene regulation. We also examined H2Bac changes in gene bodies and found a significant H2Bac decrease in THY-Tau22 mice but no change after CSP-TTK21 treatment (Fig EV4B). Significant changes, though mild, were observed on a class of highly expressed genes (included in the fourth quartile), such as the *neuropilin-2 isoform 1 precursor* gene (*Nrp2*) for example (Fig EV4C), that presents also H3K37ac enrichment on their gene bodies. Lastly, we identified significant changes at the *c-fos* genomic locus (Fig 7F). Decreased H2Bac levels were observed on distal and proximal *c-fos* enhancers (Joo *et al*, 2016) in THY-Tau22 compared to WT mice, whereas they were significantly enriched in H2Bac after CSP-TTK21 treatment, in agreement with increased *c-fos* expression levels in response to learning.

Thus, the dorsal hippocampus of THY-Tau22 mice displays decreased H2Bac levels, including at TSS, gene bodies, and CBP enhancers of highly expressed genes. Our results further indicate that CBP/p300 HAT activation with CSP-TTK21 restores an H2Bac epigenetic landscape that permits, either directly or indirectly, proper regulations of the transcriptome during learning in THY-Tau22 mice.

## Discussion

Here, we show that the CBP/p300 acetyltransferase activator CSP-TTK21 restores neuronal activity, plasticity, and memory in an AD-like Tau pathology mouse model. The hippocampus of THY-Tau22 mice was characterized by a wide decrease in H2Bac levels. CSP-TTK21 treatment globally improved H2B acetylation and partly restored the transcriptome during learning in the hippocampus of tauopathic mice, which was sufficient to reinstate plasticity (LTD, dendritic spine formation) and long-term memory processes. This study is the first to identify an epigenetic signature associated with altered neuronal plasticity in an AD mouse model. It is also the first to demonstrate reversibility at the epigenomic level with a drug targeting CBP/p300 HAT's function. Our work opens up new therapeutic options toward

the development of drugs modulating the epigenome, with HAT activators as a potent alternative to HDAC inhibitors (HDACi).

The mode of action of epigenetic modifiers in the nervous system has been poorly investigated at the genome-wide level. A first study used the HDACi trichostatin (TSA) in a physiological context: TSA treatment showed a modest impact on basal hippocampal gene expression and poorly affected the induction of IEGs by neuronal activity (Lopez-Atalaya *et al*, 2013). TSA treatment preferentially targeted already acetylated histone motifs and produced an homeostatic response that likely prevented toxic effects induced by chromatin hyperacetylation. By contrast, in the pathological conditions of tauopathy, in which a decreased acetylation is present in basal conditions, CSP-TTK21 treatment promoted general H2B histone re-acetylation, allowing to partly restore the transcriptome during learning, including induction of IEGs. Early studies demonstrated that HDACi treatment rapidly enhanced histone acetylation at *Fos* and *Jun* loci, but resulted in a second step in their transcriptional repression (Hazzalin & Mahadevan, 2005). In fact, transcriptional activation requires the recruitment of both HAT and HDAC active complexes, suggesting that the turnover of histone acetylation—as opposed to the steady-state acetylation status—is important to produce gene induction (Hazzalin & Mahadevan, 2005). Hence, treatment with a HAT activator that does not inhibit HDAC activity may here be a better therapeutic option to support neuronal activity. Another study investigating a mouse model of aging showed that the HDAC inhibitor SAHA restored H4K12ac levels and exon misusage in hippocampal tissues (Benito *et al*, 2015). However, differential splicing was not affected by amyloid pathology in the brain of APPxPS1 mice, in which SAHA also had beneficial effects, and the exact mechanisms underlying SAHA rescue, H4K12ac deregulations, and amyloid pathology were not established (Benito *et al*, 2015). Thus, the exact mode of action of HDACi and their effects in the brain remain elusive.

CBP is a transcriptional co-activator of neuronal gene expression that acts through cAMP/CREB and calcium-dependent signaling (Hardingham *et al*, 1999; Kim *et al*, 2010). A previous study demonstrated that specific CBP delivery in the hippocampus of an AD mouse model by gene transfer improved learning and memory functions (Caccamo *et al*, 2010). Pharmacological CBP/p300 acetyltransferase activation using the CSP-TTK21 molecule showed spatial memory reinforcement in normal mice (Chatterjee *et al*, 2013).

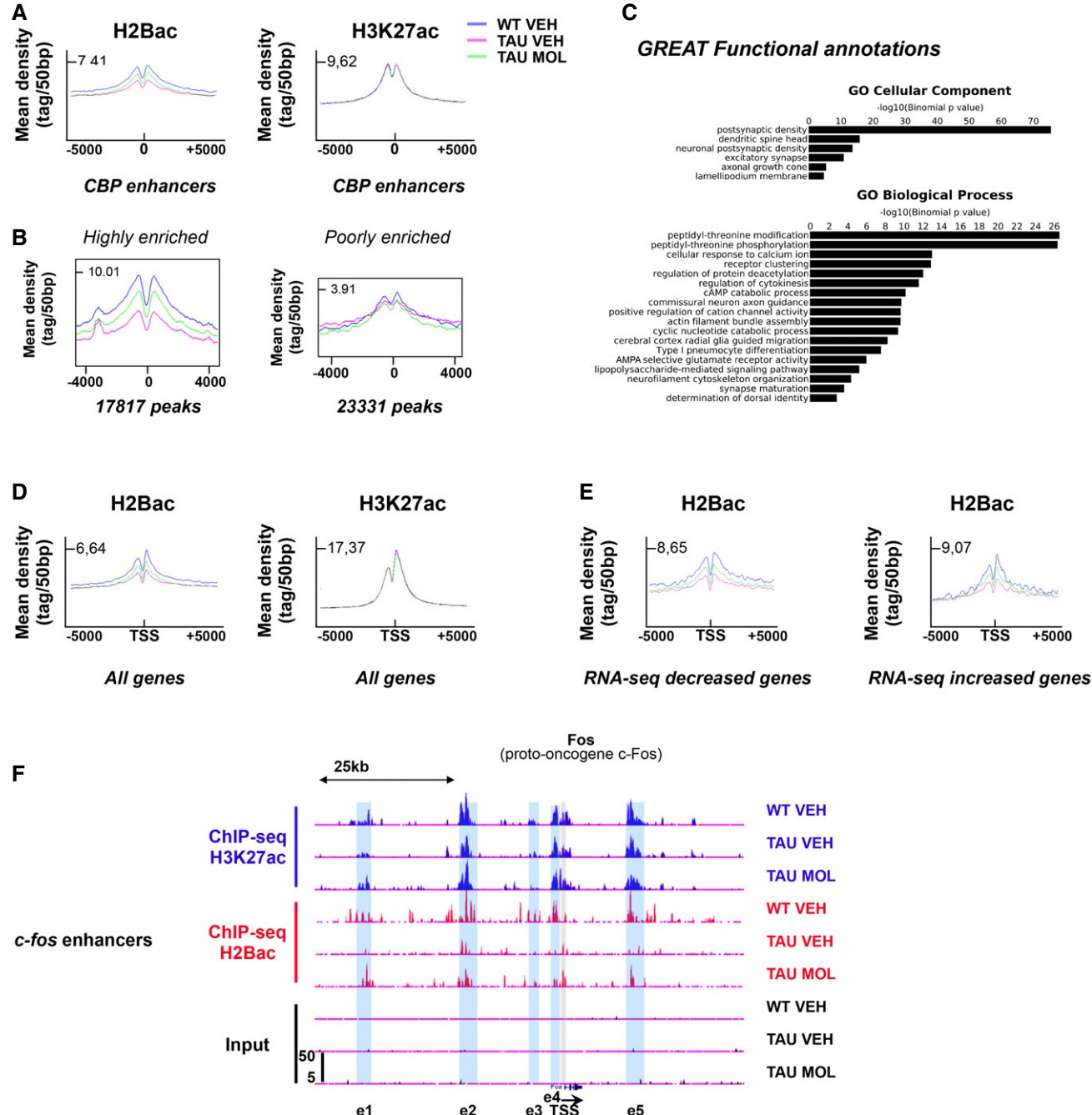

**Figure 7.  CSP-TTK21 treatment of THY-Tau22 mice affects the regulations of H2Bac enrichment at different genomic loci.**

A  Mean enrichment profiles performed with SeqMiner presenting the genomic distribution of H2Bac and H3K27ac reads obtained in ChIP-seq experiments in the different experimental groups, established along putative neuronal CBP enhancers (±4 Kb; Kim *et al*, 2010). In the dorsal hippocampus, H2Bac levels are enriched at CBP enhancers in WT WEH (blue), decreased in tauopathic mice (TAU VEH, pink), and induced by CSP-TTK21 treatment (TAU MOL, green).

B  Mean profiles were separated in two sets of loci, including 17,817 peaks highly enriched in H2Bac and showing a differential regulation in the experimental groups and 23,331 peaks that were poorly enriched in H2Bac and not regulated.

C  Functional annotation of the 17,817 peaks highly enriched in H2Bac performed with GREAT, GOTERM Cellular Component and Biological Process.

D  Mean profiles established with SeqMiner for all genes at TSS for H2Bac (Left) and H3K27ac (Right). H2Bac but not H3K24ac presents a differential regulation at TSS.

E  Mean H2Bac profiles established with SeqMiner on the differentially regulated genes obtained in transcriptomic experiments (RNA-seq), either decreased (left) or increased (right). The "basal" H2Bac TSS enrichment state of these genes is differentially regulated whether the genes are decreased or increased in response to learning.

F  UCSC genome browser view of the *c-fos* genomic locus with H3K27ac ChIP-seq (blue), H2Bac ChIP-seq (green), and input (black) signals. Blue shading indicates locations of the *c-fos* enhancers (Joo *et al*, 2016); gray shading indicates location of the promoter. Differential peaks were analyzed with SICER. This set of data shows that H2Bac levels were significantly decreased at e1 (FDR = $1.95 \times 10^{-08}$), e2 (FDR = 0.0005), and e5 (FDR = $2.44 \times 10^{-05}$) enhancers in THY-Tau22 compared to WT mice, and significantly enriched at e1 (FDR = $4.01 \times 10^{-08}$) the regions encompassing e3, e4 and TSS (FDR = $7.53 \times 10^{-07}$), and e5 (FDR = 0.001) after CSP-TTK21 treatment.

Herein, we provide for the first time a proof of principle that a HAT activator molecule is able to restore plasticity and memory functions in a pathological brain. Treatment of THY-Tau22 mice with CSP-TTK21 prior being challenged to spatial training promoted the upregulation of genes enriched in the cAMP signaling pathway, including IEGs, and in ion channel/transporter activity functions, in agreement with a role of CBP/p300 in the regulation of ion homeostasis and neuronal excitability. Importantly, CSP-TTK21 counterbalanced the transcription of downregulated genes in THY-Tau22 mice that have been previously associated with improved cognitive functions: *Klotho* and *Neurotensin*, a result that we could validate at the protein level. Klotho was first described to counteract aging, and more recently, showed beneficial effects in a mouse model of AD (Kurosu *et al*, 2005; Kuang *et al*, 2017; Leon *et al*, 2017; Massó *et al*, 2017). The neurotensin system is altered in the temporal lobe of AD patients (Gahete *et al*, 2010), and APP/PS1 mice microinjected with Neurotensin show improved spatial memory functions and neuronal excitability (Xiao *et al*, 2014). Altogether, these data support a mechanistic model in which activation of the acetyltransferase CBP in a diseased brain can re-establish functional neuronal networks through induced expression of genes involved in neuronal stimulation (IEGs, cAMP pathway, ion homeostasis) and important cognitive enhancers.

An important finding is that we established an epigenetic signature associated with H2Bac in the hippocampus of cognitively deficient tauopathic mice. Very few studies investigated histone acetylation in pathological context at the genome-wide level. Our recent studies showed selective decreased H3K27ac at neuronal identity genes regulated by a super-enhancer, in the striatum of Huntington's disease mice (Achour *et al*, 2015; Le Gras *et al*, 2017). In the hippocampus of the CK-p25 AD mouse model, H3K27ac levels were decreased at promoter/enhancers of genes associated with synapse and learning functions and preferentially bound by CBP (Gjoneska *et al*, 2015). Our data show for the first time a significant alteration of H2Bac levels in mice bearing AD Tau hallmarks. As H3K27ac levels between THY-Tau22 and WT mice were unchanged, we suggest that H2Bac loss may be an early mark of a disease state in the hippocampus.

Few studies have investigated H2Bac genomic distribution in normal tissue. A previous study reported that H2Bac profile mostly covers intragenic regions and promoters in hippocampal tissues (Lopez-Atalaya *et al*, 2013). They also found H2Bas enrichment at CBP enhancers, also defining H2B as a preferred target mark for CBP (Alarcon *et al*, 2004; Valor *et al*, 2011). Here, we confirm this genomic distribution and precisely show that acetylation of H2B N-terminus is enriched at the TSS of the 50% mostly expressed hippocampal genes (Fig 6B). H2BK5ac has already been reported as an important modification associated with active gene transcription in other tissues/cell types (Karlić *et al*, 2010). Moreover, H2BK20ac was recently suggested to contribute to cell type–specific gene regulation and biological functions through distal and proximal cis-regulatory elements (Kumar *et al*, 2016). Thus, the decrease in H2Bac in THY-Tau22 hippocampus may have deleterious functional consequences. Yet, the severe H2Bac decrease observed in mice with a conditional knock-down of CBP (CBP cKO mice) was not sufficient to induce memory deficits (except for novel object recognition) or neuronal death (Valor *et al*, 2011). In our tauopathy model, decreased H2Bac and CBP levels are only part of the pathological process, which also includes neuroinflammation in contrast to CBPcKO mice. H2Bac loss in THY-Tau22 mice may thus synergize with other dysfunctions.

Remarkably, CSP-TTK21 treatment of THY-Tau22 mice resulted in a global increase in H2Bac acetylation at TSS, CBP enhancers (Kim *et al*, 2010). It is noteworthy that deregulated peaks (presenting a decrease in H2Bac enrichment in tauopathic mice and an increase by the treatment) were associated with genes involved mainly in GO terms related to neuronal signaling pathways—including cAMP and calcium, glutamatergic synapse, and LTP/LTD processes—all of these pathways being dysregulated in tauopathic mice (Van der Jeugd *et al*, 2011; Ahmed *et al*, 2015; Burlot *et al*, 2015). This suggests a link between the restoration of H2Bac levels in the hippocampus and that of hippocampal functions (i.e., spatial memory formation, LTD, dendritic spine formation). However, all these genes were not deregulated in terms of gene transcription in basal conditions. In addition, deregulated transcription in THY-Tau22 compared to WT mice in learning conditions also lacked correlation with H2Bac enrichment in the basal state, as they followed the same pattern of enrichment (decreased in tauopathic mice and increased by the molecule) whether they were transcriptionally up- or downregulated after learning (Fig 7E). This indicated that gene transcription *per se* is not solely dependent on H2Bac enrichment. It seems more likely that CBP/p300 HAT activity maintains a balanced H2Bac epigenomic landscape in the hippocampus, which may contribute to acetylation homeostasis necessary for neuronal functions. A lack of clear correlation between acetylation levels at specific genes and their transcriptional status has been reported previously, either in learning and memory (Lopez-Atalaya & Barco, 2014; Halder *et al*, 2016) or in response to HDAC inhibition (Lopez-Atalaya *et al*, 2013). A growing body of evidence suggests that the specific genomic topologies favor correct spatiotemporal gene expression programs in response to challenge (Mitchell *et al*, 2014; Madabhushi *et al*, 2015; Watson & Tsai, 2017) and the CBP/p300-H2Bac balance may be an important player in such mechanisms.

Lastly, how such epigenetic dysfunction relates to Tau pathologies is still an open question. Interestingly, Tau protein is found in the nucleus (Wei *et al*, 2008; Sultan *et al*, 2011; Qi *et al*, 2015), a cellular compartment where Tau levels progressively increase during aging (Gil *et al*, 2017). On the one hand, nuclear Tau physiological functions were recently proposed to participate in the maintenance of pericentric heterochromatin integrity (Mansuroglu *et al*, 2016). On the other hand, H2B overexpression was shown to deregulate heterochromatin architecture (Ito *et al*, 2014; Medrano-Fernández & Barco, 2016). Further experiments are therefore required to understand whether pathological nuclear Tau accumulation can (directly or indirectly) alter euchromatin topology, H2Bac levels, and CBP-dependent regulations.

In conclusion, this study identifies that pharmacological activation of CBP/p300 HAT function with a new epi-drug (CSP-TTK21) stands as a therapeutic option for AD-related disorders. Indeed, CSP-TTK21 treatment restored plasticity in the diseased brain of mice presenting an AD-like Tau pathology. One reason why most AD clinical studies may have failed is that cognitive dysfunctions associated with AD patients emerge late in the progression of the disease, when the brain is already affected by amyloid β deposition, neurofibrillary tangles, and cell death (Hyman *et al*, 2012). The possibility to reinstate some plasticity in affected brains by activating the acetyltransferase function of CBP/p300 could lead to more successful therapeutic options that may delay the cognitive decline and improve the condition of AD patients.

# Materials and Methods

### Animals

Heterozygous THY-Tau22 transgenic mice were bred on a C57BL6/J background. THY-Tau22 mice overexpressed mutated human Tau protein (G272V and P301S) under the control of a Thy1.2 promoter allowing a specific neuron expression that starts at postnatal day 6 (Schindowski *et al*, 2006). All mice used in this study were 8-month-old male mice, except in Fig 1A and B in which we used 12-month-old male mice and in one H2Bac ChIP-seq replicate, where we used 8-month-old female mice. All animals were kept in standard animal cages under conventional laboratory conditions (12-h/12-h light–dark cycle, temperature: 22°C ±2, humidity: 55% ±5) with *ad libitum* access to food and water. All behavior experiments were conducted between 9:00 and 12:00 am. Experimental protocols and animal care were in compliance with the institutional guidelines (council directive 87/848, October 19, 1987, Ministère de l'agriculture et de la Forêt, Service Vétérinaire de la Santé et de la Protection Animale) and international laws (directive 2010/63/UE, February 13, 2013, European Community) and policies (personal authorizations #67-117 for A.-L.B., and #I-67UnivLouisPasteur-F1-04 for R.C.). Our project has been reviewed and approved by the ethics committee of Strasbourg, France (#AL/100/107/02/13 and APAFIS#5118-2016042017216897v9).

### Drug production, testing, and treatment

See Appendix Supplementary Methods and Appendix Fig S2, and also Chatterjee *et al* (2013).

### Morris water maze evaluations

Atlantis MWM tank was placed in a room with several visual extra maze cues. Water opacified with powdered chalk (Blanc de meudon) was maintained at 21°C. Mice were habituated to the set-up for two consecutive days (*habituation 1 and 2*). During habituation 1, mice were allowed to discover the pool filled with 5 cm height of water and a visible platform during 60 s. During habituation 2, mice were allowed to swim in the pool filled with water and without a platform for 60 s. Mice were trained for the next 5 days (*acquisition 1–5*) to locate a hidden platform beneath the water using spatial cues present in the room. Each acquisition day consisted of four trials of maximum duration of 60 s in each trial. Each trial concluded immediately after mice reached the platform or after the completion of 60 s. Mice failing to find the platform were gently guided to reach the platform and was allowed to stay for 8–10 s. Probe test was performed 10 days after the last training day. During the probe test, the platform was removed and mice were placed at the center of the pool and allowed to swim for 60 s. The different parameters (distance, time, etc.) were recorded by a video tracking system (ANY-maze). The experimenter was blind to genotype and treatment. The mice were uniquely identified according to an identification chip. Mice were randomly distributed as their order of passage in the Morris water maze during the different days of training (so that they do not perform the tap.22.sk always at a given time). Genotype and treatment were uncovered before analyses.

### Statistics

For behavioral data, we have used a sample size to more than 10. Exclusion/inclusion criteria: Outliers were defined as results differing from the mean by 3 SD. Stressed mice were defined upon ethical criteria: stayed immobile in the cage or showed aggressive behavior. A floating behavior in the Morris water maze was defined as a floating during more than 30s/assay of 60s. These mice were excluded from the experiment and usually represented on average 1 per experiment. One- or two-way ANOVA was used for data analysis with repeated measures when appropriate. Escape latencies recorded during acquisition were analyzed using ANOVA, followed by Newman–Keuls multiple-comparisons test when appropriate. For the probe tests, the time spent in the target quadrant was compared with the chance level (15 s for a test duration of 60 s in four quadrants) using a one-sample Student's *t*-test. ANOVA factorial was used to analyze the "treatment" and "genotype" effect for the time spent in the target quadrants and the average time in the other three quadrants. Results were expressed as means of +SEM. Values of $P < 0.05$ were considered as statistically significant.

### Immunohistochemistry

For heat-induced epitope retrieval, floating sections were kept in sodium citrate (pH 6, 10 mM, 37°C, 30 min). After permeabilization (PBS1X/Triton X-100 2%, 15 min), unspecific labeling was blocked (PBS1X/Triton X-100 0.1%/horse serum 5%, 1 h at RT) and slices were incubated overnight with polyclonal anti-CBP antibody (1/50; ab2832; Abcam), washed, and further incubated with anti-rabbit Alexa Fluor 594 (red, 1/1,000) antibody (Invitrogen, Thermo Fisher Scientific) for 1 h. IHC was performed sequentially and the second antibody was added in a second step. Slices were incubated overnight with monoclonal anti phospho-Tau (Thr212, Ser214) antibody (1/1,000; AT100, MN1060 Thermo Fisher Scientific), washed, and further incubated with anti-mouse Alexa Fluor 488 (green, 1/1,000) antibody (Invitrogen, Thermo Fisher Scientific). Slices were incubated with the Hoechst dye 33342 (1 mg/ml; 5 min) and mounted in Mowiol for observation. Acquisitions were performed using a fluorescence microscope coupled with an ApoTome module (Zeiss). Photomicrographs were captured using the z-stack mode with 0.5 μm of thickness between slices, 18 in-depth slices were taken and flattened using the maximum intensity projection (MIP).

### Biochemical analyses

Mice were either left in their home cage (basal conditions) or trained for three consecutive acquisition days for spatial reference memory in the Morris water maze and left in their home cage for 1 h after the last trial (learning conditions). Mice were killed by cervical dislocation, the brains were extracted, and the dorsal hippocampus was immediately dissected out, flash-frozen in liquid nitrogen, and stored at −80°C. The time at which the mice were killed was randomized according to their experimental group and was consistently performed between 9:00 and 11:00 am.

#### Western blot analyses

Dorsal hippocampi were finely chopped on ice using a razor blade and homogenized in Laemmli buffer with a disposable pestle in microcentrifuge tubes. Tissues were sonicated for 15 s twice

(ultrasonic processor, power 40%) followed by heating at 70°C for 10 min. Lysates were centrifuged at 20,000 g for 5 min, and supernatant was stored at −80°C until use. Estimation of the protein concentration in the lysate was determined by using RC-DC Protein Assay (Bio-Rad) kit. Proteins were loaded on Midi-PROTEAN TGX Stain-Free™ Precast Gels (26 wells, 4–20%, Bio-Rad). The use of stain-free imaging allows for the normalization of bands to the total protein on a blot. A UV-induced 1-min reaction of the gels after protein migration produces fluorescence. The fluorophores remain covalently bound to the proteins throughout blotting and may be subsequently visualized in gels or on the nitrocellulose membranes for validation in the Western blotting workflow. Antibodies used for Western blots were acetyl-histone H2BK5 (1/1,000; #07-382; Millipore), acetyl-histone H2BK5K12K15K20 (1/1,000; #07-373; Millipore), H3 histone (1/2,000; #ab1791; Abcam), acetyl-histone H3K9K14 (1/1,000; #06-599; Millipore), histone H2B (1/4,000; #H2-28; Euromedex), acetyl-histone H3K27 (1/2,000; #ab4729; Abcam), acetyl-histone H4 (1/4,000; #39165, Active motif), Phospho-Tau (Ser404) (1/1,000; #44-758G, Thermo Fisher Scientific), Tau 5 (1/1,000; #AHB-0042, Invitrogen), CBP (1/500; #sc-369, Santa Cruz Biotechnology), GFAP (1/2,000; #MAB360; Millipore), Neurotensin (1/1,000; #BML-NA1230; Enzo Life Sciences); Klotho (1/5,000; #AF1819,R&D Systems), actin (1/1,000; #A2066; Sigma). HRP conjugated secondary antibodies against mouse and rabbit (Jackson ImmunoResearch). Blots were revealed with ECL (Clarity, Bio-Rad) with a ChemiDoc Touch system (Bio-Rad). Results were quantified using ImageLab software.

### RNA extractions

Total RNA was extracted from the dorsal hippocampal tissues using TRIzol reagent (Invitrogen). Freshly dissected tissues were chopped, homogenized in 400 μl of TRIzol reagent, and frozen (20 min at −80°C), followed by 3-min centrifugation at 14,000 g before chloroform/isoamyl extraction. DNA precipitation was performed on the supernatant using isopropanol and RNase-free glycogen (10 min at RT). The pellet was washed once with 70% ethanol and resuspended in Milli-Q water for DNAse treatment (30 min at 37°C). RNA samples were further purified using a phenol/chloroform extraction, followed by a new precipitation (overnight at −20°C, with 100% ethanol, 3 M NaOAC). After two last 70% ethanol washes, pellets were air-dried and resuspended in 30 μl nuclease-free Milli-Q water, heated 6 min at 50°C, and RNA concentration estimation was performed.

### Real-time qRT–PCR analyses

Samples with $OD^{260/280}$ and $OD^{260/230}$ ratio close to 2.0 were considered for cDNA synthesis. 0.5–1 μg of total RNA was reverse-transcribed (iScript cDNA synthesis kit; Bio-Rad). Bio-Rad iCycler System (CFX) was used to perform qPCR using SsoAdvanced SYBR Green SuperMix (Bio-Rad). A specific standard curve was performed in parallel for each gene, and technical duplicates were performed. The conditions for qRT–PCR were 3 min at 94°C, followed by 40 cycles of 45 s at 60°C and 10 s at 94°C. Data were analyzed by gene regression using iCycler software and normalized to housekeeping genes (RNA polymerase II, 36B4) levels.

### Statistics for biochemical studies

For these studies, we have tried to always have $n$ values > 5. For Western blot studies comparing WT to THY-Tau22 mice, inclusion

was verified by performing an immunostaining against a pathological form of Tau protein (Tau5, Tau-Ph404). All samples were used until exhaustion which explains some variability in the number of Western blot samples ($n$ = 5–7). For RT–qPCR validation experiments, all samples ($n$ = 5) were included. All statistical analysis involving three groups (WT, THY-Tau22 CSP, and THY-Tau22 CSP-TTK21) were performed using ANOVA followed by uncorrected Fisher's for multiple-comparisons tests. For comparison between two groups, Student's $t$-test was performed. Data were expressed as the mean + SEM. The significant level was set at $P < 0.05$.

### Dendritic spines labeling and counting after GFP staining

*Stereotaxic injections of lentivirus-GFP*
See Appendix Supplementary Methods.

*Immunohistochemistry*
Mice were deeply anesthetized with pentobarbital (50 mg/kg) and transcardially perfused with PB (0.1 M phosphate buffer) followed by cold 4% paraformaldehyde in PB (PFA). Brains were removed and fixed overnight in 4% PFA and subsequently merged in 30% sucrose solution at 4°C until the brain sank. Brains were cut on a freezing microtome into 50-μm-thick coronal sections covering the whole hippocampus. Sections were kept in cryoprotectant [0.1 M PB, 0.15 M NaCl, 30% (v/v) ethylene glycol, 30% (v/v) glycerol] at −20°C until immunostaining. For immunostainings, free-floating brain sections were extensively rinsed in PBS-T (0.25% Triton X-100 in PBS; 137 mM NaCl, 2.7 mM KCl, 10 mM $Na_2HPO_4$, 1.8 mM $KH_2PO_4$) and incubated overnight in the following primary antibodies at 4°C: goat anti-GFP (Rockland, 1:500) and mouse anti-NeuN (Millipore, 1:1,000) diluted in PBS-T. Brain sections were rinsed and incubated for 1 h at room temperature with, respectively, donkey anti-goat Alexa 488-conjugated and donkey anti-mouse Alexa 555-conjugated (Life Technologies; all diluted 1:500 in PBS-T). After several rinses, brain sections were mounted in Hoechst mixed in Mowiol (Life Technologies; 1:10,000), coverslipped, and kept at 4°C. Images of basal dendritic segments of GFP-labeled CA1 pyramidal neurons were acquired using a confocal microscope (Leica TCS SP8; Leica, Heidelberg, Germany) equipped with 63× HCX PL APO (1.40 NA) objective (Toulouse Réseau Imagerie, FRBT-CBD). The laser power, photomultiplier gain, and offset were kept constant for every image. Four images of 1,024 × 1,024 pixels were acquired at 8,000 Hz scanning speed and averaged to increase the signal-to-noise ratio. Dendritic spine images were acquired in z-series at 0.19-μm interval. The voxel size was 30.06 nm in x–y directions and 193.81 nm in z-direction. Stack images of 12-bit files were deconvolved with Huygens Essential deconvolution software (SVI) with five iterations and images were imported into Imaris XT (Bitplane AG) for analysis.

### Dendritic spines labeling and counting after Golgi staining

Mice that had underwent a 4-day training in the MWM were killed 4 days later. The Rapid Golgi stain kit (FD Neurotechnologies, Inc.) was used according to the manufacturer's instructions. Freshly dissected brains were immersed in solution A and B for 2 weeks at room temperature and were then transferred to solution C for

another 24 h at 4°C. 150-μm-thick coronal sections containing the rostro-caudal axis of hippocampal region CA1 were made using a Vibratome (VT1000M; Leica). Images were acquired using a bright-field microscope equipped with automated motorized stage and MorphoStrider software (Explora Nova, La Rochelle, France). 63× magnification images were taken from secondary and tertiary dendrites from CA1 pyramidal neurons. For each genotype, condition and treatment 8–10 dendritic segments per animal were analyzed.

## Quantification of dendritic spines

Dendritic spines were defined as protrusions from the dendritic shaft and identified based on their morphological appearance. Spines were classified into three different types according to Harris et al (1992): head spines (protrusion with neck and head), stubby (protrusion with no obvious neck or head), and filopodia (protrusion with long neck and no head). For each animal, 24–35 dendritic segments were analyzed from 5 to 10 neurons. The spine density was presented as the number of spines per 1 μm of dendritic length.

### Statistics
Dendritic spines counting scores were from about 25 (Golgi staining) and 70 (GFP lenti) dendritic segments which were from material of 15–30 neurons and two to three animals/condition. Data are presented as mean ± SEM. For GFP staining, Kolmogorov–Smirnov test was performed on each data set to test the distribution normality. Statistical analyses were run with Prism software (GraphPad 5.0). Dendritic spine analyses were compared across groups by means of one-way ANOVA followed by post hoc analyses that were performed using Holm–Sidak multiple-comparisons tests. For Golgi staining, dendritic spines were compared across different groups by using a two-way ANOVA followed by Newman–Keuls multiple-comparisons tests.

## Ex vivo electrophysiology (LTD)

Mice were anesthetized with halothane and decapitated. The brain was rapidly removed from the skull and placed in chilled (0–3°C) artificial cerebrospinal fluid (ACSF) containing 124 mM NaCl, 3.5 mM KCl, 1.5 mM $MgSO_4$, 2.5 mM $CaCl_2$, 26.2 mM $NaHCO_3$, 1.2 mM $NaH_2PO_4$, and 11 mM glucose. Transverse slices (300–400 μm thick) were cut with a tissue chopper and placed in ACSF in a holding chamber, at 27°C, for at least 1 h before recording. Each slice was individually transferred to a submersion-type recording chamber and submerged in ACSF continuously superfused and equilibrated with 95% $O_2$, 5% $CO_2$. For electrophysiological recordings, a single slice was placed in the recording chamber, submerged, and continuously superfused with gassed (95% $O_2$, 5% $CO_2$) ACSF (28–31°C) at a constant rate (2 ml/min). Extracellular field excitatory postsynaptic potential (fEPSPs) were recorded from the apical dendritic layer of the CA1 area, using a glass micropipette filled with 2 M NaCl. fEPSPs were evoked by the electric stimulation of Schaffer collaterals/commissural pathway at 0.1 Hz with a bipolar tungsten stimulating electrode placed in the stratum radiatum (100 μs duration). The averaged slope of three fEPSPs was measured using p-clamp software (Axon Instruments) for 20 min before conditioning stimulation. Long-term depression (LTD) was induced by applying a low-frequency stimulation at 2 Hz (1,200 pulses for 10 min). The sequence was repeated three times, with an interburst interval of 10 s. Testing with a single pulse was resumed for 60 min to determine the level of LTD. Any change in synaptic strength was expressed relative to the control level (100%).

### Statistics
Average values of the fEPSP slope were calculated for the last 10 min and expressed as a percentage of the baseline response (% baseline) ± SEM. In order to take into account the correlations inherent to repeated measures in electrophysiological recordings, P-values were calculated using multivariate analyses of variance followed by post hoc unpaired Student's t-tests using StatView software. In all cases, differences were considered significant when $P \leq 0.05$.

## RNA libraries and sequencing

RNA-seq libraries (n = 3/group) were generated from 300 ng of total RNA Illumina® TruSeq® RNA Sample Preparation Kit v2 (Part Number RS-122-2001). Briefly, following purification with poly-T oligo-attached magnetic beads, the mRNA was fragmented using divalent cations at 94°C for 2 min. The cleaved RNA fragments were copied into first-strand cDNA using reverse transcriptase and random primers. Second-strand cDNA were synthsized using DNA Polymerase I and RNase H. Following the addition of a single "A" base and subsequent ligation of the adapter on double-stranded cDNA fragments, the products were purified and enriched with PCR [30 s at 98°C; (10 s at 98°C, 30 s at 60°C, 30 s at 72°C) × 12 cycles; 5 min at 72°C] to create the cDNA library. Surplus PCR primers were further removed by purification using AMPure XP beads (Beckman Coulter), and the final cDNA libraries were checked for quality and quantified using capillary electrophoresis. Sequencing was performed on the Illumina Genome Hiseq2500 as single-end 50 base reads following Illumina's instructions.

### RNA-sequencing analyses and statistics
Reads were mapped onto the mm9 assembly of mouse genome using TopHat v2.0.10 (Kim et al, 2013) and the bowtie v2.1.0 aligner. Gene expression was quantified from uniquely aligned reads using HTSeq v0.5.4p3 (Anders et al, 2013) and gene annotations from Ensembl release 67. Read counts were normalized across libraries with the method proposed by Anders and Huber (2010). Statistical analysis was performed with the method proposed by Love et al (2014) implemented in the DESeq2 Bioconductor library (v1.0.19). Resulting P-values were adjusted for multiple testing by using the Benjamini and Hochberg (1995) method. Heatmaps were performed with R (heatmap.2 function, Euclidean distance, and complete linkage were used to perform gene clustering). Heatmap representing expression z-scores calculated as row Z score = [(Value) – Mean (Row)]/ [Standard deviation (Row)]. Cross-comparison of RNA-seq data was performed using Venny 2.1 (Oliveros, 2007–2015). Gene ontology (GO) analyses for functional enrichments were performed using the tools DAVID (da Huang et al, 2009) and/or GREAT (McLean et al, 2010). For analyses with GREAT, defaults setting were used. Whole Mouse genome (mm9) was used as background. Top-enriched terms are shown (P-values < 0.05 were considered).

For RNA-seq analyses presented in Fig 3, three out of 12 samples were excluded because a great variability was seen in the heatmaps

quantifying the global RNA-seq sample differences, likely due to genetic derivation; of note, these three mice—that had been randomly assigned to an experimental condition—originated from the same litter out of five litters total. These criteria were decided *a posteriori*.

### ChIP experiments

Freshly dissected tissues were chopped with a razor blade and rapidly put in 1.5 ml PBS containing 1% formaldehyde for 10 min at room temperature (RT) followed by the addition of glycine (0.125 M final concentration). Dorsal hippocampi from eight mice were pooled per sample. Tissue samples were then processed as in Bousiges *et al* (2010) and sonicated with a Diagenode Bioruptor (35 cycles with 30-s ON/30-s OFF on High Power). Sonicated chromatin was centrifuged 10 min at 14,000 *g*, and the supernatant was collected and diluted 10 times in ChIP dilution buffer (0.01% SDS, 1.1% Triton X-100, 1.2 mM EDTA, 16.7 mM Tris-Cl, pH 8.1, 167 mM NaCl). A fraction of supernatant (50 μl) from each sample was saved before IP for "total input chromatin". Supernatants were incubated overnight (4°C) with 1/1,000 primary antibody [H2BK12K15ac (ab1759), H2BK5ac (07-382); H3K27ac (ab4729)], followed by protein A Dynabeads (Invitrogen) for 2 h at RT. After several wash steps (with low salt, high salt, LiCl, and TE buffers), complexes were eluted in 300 μl elution buffer (1% SDS, 0.1 M NaHCO$_3$). The crosslinking was reversed (overnight at 65°C), and the DNA was subsequently purified with RNase (30 min, 37°C), proteinase K (2 h, 45°C), phenol/chloroform, and chloroform/isoamyl extractions. After overnight ethanol precipitation (at −20°C), two last 70% ethanol washes were done before air drying the resulting DNA pellet. The pellets were resuspended in 30 μl nuclease-free Milli-Q water, and the supernatant was transferred in low binding tubes and brought to the sequencing platform for DNA concentration estimation (Qubit fluorometric quantitation).

### ChIP-seq libraries

ChIP-seq libraries were prepared using NEXTflex ChIP-seq Kit (#5143-02, Bioo Scientific) following the manufacturer's protocol (V12.10) with some modifications. Briefly, 10 ng of ChIP-enriched DNA or input DNA was end-repaired using T4 DNA polymerase, Klenow DNA polymerase, and T4 PNK, then size-selected, and cleaned-up using Agencourt AMPure XP beads (#A63881, Beckman). A single "A" nucleotide was added to the 3′ ends of the blunt DNA fragments with a Klenow fragment (3′ to 5′exo minus). The ends of the DNA fragments were ligated to double-stranded barcoded DNA adapters (NEXTflex ChIP-seq Barcodes-6, #514120, Bioo Scientific) using T4 DNA Ligase. The ligated products were enriched by PCR [2 min at 98°C; (30 s at 98°C, 30 s at 65°C, 60 s at 72°C) × 14 cycles; 4 min at 72°C] and cleaned-up using Agencourt AMPure XP beads. Prior to analyses, DNA libraries were checked for quality and quantified using a 2100 Bioanalyzer (Agilent). The libraries were loaded in the flowcell at 8 pM concentration, and clusters were generated using the Cbot and sequenced on the Illumina Genome Hiseq2500 as single-end 50 base reads following Illumina's instructions. Image analysis and base calling were performed using RTA and CASAVA.

### ChIP-sequencing analyses

Reads were mapped onto Mouse reference assembly mm9/NCBI37 using bowtie v0.12.7 aligner (Langmead *et al*, 2009). Peak detection

### The paper explained

**Problem**

Alzheimer's disease (AD) is a neurodegenerative disease first affecting memory functions and progressively leading to massive neuronal loss and dementia. There is currently no cure, and some recent clinical trials failed.

**Results**

Chromatin acetylation, a critical regulator of synaptic plasticity and memory processes, is thought to be altered in neurodegenerative diseases. Our study provides evidence of acetylation dysregulations in an AD-like Tau pathology mouse model. It further brings an *in vivo* proof of concept that a treatment with a CBP/p300 acetyltransferase activator molecule (CSP-TTK21) efficiently restores neuronal activity, plasticity, and memory in this mouse model, together with molecular evidence of rescued acetylation in the brain.

**Impact**

The possibility to reinstate some plasticity in affected brains by activating the acetyltransferase function of CBP/p300 could lead to more successful therapeutic options that may delay the cognitive decline and improve the condition of AD patients.

of H2Bac and H3K27ac was performed using SICER v1.1 (Zang *et al*, 2009) with the following parameters: window size: 200; gap size: 1,000 for H2Bac; and 600 for H3K27Ac. Peaks were annotated relative to genomic features using Homer (Heinz *et al*, 2010) with annotation from Ensembl v67. Differential peaks were analyzed using SICER with the SICER-df.sh script. Global comparison of samples and clustering analysis were performed using seqMINER (Ye *et al*, 2011). As reference coordinates, we used RefSeq genes for Mouse mm9 genome. Increased and decreased regions were selected if their *P*-values < $10^{-5}$. Biological duplicates were performed for the H2Bac mark. They were analyzed independently in the same manner, and only differential peaks replicating were further analyzed (Fig EV3A and B). Inputs were used as controls.

Integrated analysis of RNA-seq and ChIP-seq data was done using the open Galaxy platform GalaxEast (http://www.galaxeast.fr).

Nucleotide coverage between H2Bac and H3K27ac was calculated using bedtools jaccard (bedtool Fisher exact test, *P*-value = 0).

### Normalization of ChIP-seq data

ChIP-seq data were normalized based on a method that consists of equalizing background regions. Briefly, read counts were computed in all regions of 5,000 nt in length of the mouse genome (mm9 assembly) for each of the samples to normalize. Read counts were then compared across samples and used to estimate a scaling ratio. Finally, the scaling ratio was used to remove randomly chosen reads in the samples to be normalized.

## Data availability

The datasets produced in this study are available in the following databases:

- RNA-seq data: Gene Expression Omnibus GSE103359 https://www.ncbi.nlm.nih.gov/geo/query/acc.cgi?acc=GSE103359

• ChIP-seq data: Gene Expression Omnibus GSE103358 https://www.ncbi.nlm.nih.gov/geo/query/acc.cgi?acc=GSE103358

Expanded View for this article is available online.

## Acknowledgements

The authors thank Dr. Matthieu Gerard (CEA Saclay, Gif-sur-Yvette, France) for his helpful comments on an early version of the manuscript. The authors are grateful to O Bildstein, O Egesi, G Edomwonyi, and C Strittmatter (UMR 7364) for their assistance in animal care. We also thank Drs Nathalie Mandairon and Marion Richard (CRNL, Lyon, France) for their counseling on lentiviruses, and Gisèle Froment, Didier Nègre, and Caroline Costa from the lentivector production facility/SFR BioSciences de Lyon (UMS3444/US8). Sequencing was performed by the GenomEast Platform, a member of the "France Génomique" consortium (ANR-10-INBS-0009). TKK is a recipient of the Sir JC Bose national fellowship (Department of Science and Technology, Govt. of India). SC was supported by IFCPAR/CEFIPRA (No. 4803-3). RC was a recipient of a doctoral fellowship from the French government. AS-A was supported by the ANR (ANR-12-MALZ-0002-01). LB and DB are supported by programs d'investissements d'avenir LabEx (excellence laboratory) and the DISTALZ (Development of Innovative Strategies for a Transdisciplinary approach to ALZheimer's disease), France Alzheimer, FHU VasCog research network (Lille, France), Fondation pour la Recherche Médicale, LECMA/Alzheimer Forschung Initiative, Fondation Plan Alzheimer, Inserm, CNRS, Université Lille 2, Lille Métropole Communauté Urbaine, Région Nord/Pas-de-Calais, FEDER, DN2M, and FUI MEDIALZ. This work was supported by the CNRS, the University of Strasbourg, ANR (ANR-12-MALZ-0002-01 to ALB), France Alzheimer (to ALB), the Department of Biotechnology, the Government of India (Grant/DBT/CSH/GIA/1752 to TKK), the Jawaharlal Nehru Centre for Advanced Scientific Research (JNCASR), and the Indo-French Centre for the Promotion of Advanced Research (IFCPAR/CEFIPRA (No. 4803-3 to TKK and ALB)). We warmly thank Alsace Alzheimer 67 and Ligue Européenne Contre la Maladie d'Alzheimer for supporting earlier work of this study.

## Author contributions

A-LB designed the work. A-LB, KM, SC, AS-A, RC, BC, LT, OB, SLG, CK, PD, PP, CR, SHS, MK, PC, and ME performed the experiments. A-LB, TKK, KM, SC, AS-A, RC, and J-CC analyzed the data. Reagents (CSP, CSP-TTK21) production and quality control was supervised by TKK; A-LB, KM, J-CC, and DB wrote the initial draft; and A-LB, KM, J-CC, DB, TKK, SC, AS-A, RC, SLG, CK, CR, PD, and LB reviewed and edited the manuscript. All authors read and approved the final manuscript.

## Conflict of interest

The authors declare that they have no conflict of interest.

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
