## [Review Process File · EMBO Molecular Medicine]

Reinstating plasticity and memory in a tauopathy mouse model with an acetyltransferase activator

Snehajyoti Chatterjee, Raphaelle Cassel, Anne Schneider-Anthony, Karine Merienne, Brigitte Cosquer, Laura Tzeplaëff, Sarmistha Halder Sinha, Manoj Kumar, Piyush Chaturbedy, Muthusamy Eswaramoorthy, Stéphanie Le Gras, Céline Keime, Olivier Bousiges, Patrick Dutar, Petnoi Petsophonsakul, Claire Rampon, Jean-Christophe Cassel, Luc Buée, David Blum, Tapas K. Kundu and Anne-Laurence Boutillier

Review timeline:

Submission date:	16 October 2017
Editorial Decision:	21 November 2017
Revision received:	17 July 2018
Editorial Decision:	13 August 2018
Revision received:	31 August 2018
Accepted:	5 September 2018

Editor: Céline Carret

Transaction Report:

1st Editorial Decision

21 November 2017

Thank you for the submission of your manuscript to EMBO Molecular Medicine. We have now heard back from the three referees whom we asked to evaluate your manuscript.

You will see that the referees found the data of interest and the studied compound with therapeutic potentials, which fits our scope and aims. However, all referees noted over-interpreted results and suggest a nice battery of further experimentation and better analyses to improve the conclusiveness of the findings. A better presentation of the findings is also suggested.

We would welcome the submission of a revised version within three months for further consideration and would like to encourage you to address all the criticisms raised as suggested to improve conclusiveness and clarity. Please note that EMBO Molecular Medicine strongly supports a single round of revision and that, as acceptance or rejection of the manuscript will depend on another round of review, your responses should be as complete as possible.

I look forward to receiving your revised manuscript.

***** Reviewer's comments *****

Referee #1 (Remarks for Author):

The authors show that treatment of TAU transgenic mice with a CBP HAT activator (CSP-TTK1) improves memory deficits and other associated correlates. They also show some changes in histone acetylation and at the transcriptional level. In particular the transcriptional study in mice that have undergone memory training is elegant and may lead to identification of the molecular mode of action of CSP-TTK1 in the future. The study as it stands has a number of experimental shortcomings and at times over-emphasizes outcomes. Furthermore, a direct connection between H2B acetylation and memory improvements in TAU mice is missing. The issues addressed below would need to be addressed in full before a revised version of the manuscript should be re-assessed.

Issues in order of appearance in the manuscript

Fig. 1A: The blots do not convincingly show a reduction in CREB. If CREB levels are indeed reduced, but its phosphorylation not, then relative CREB activity is likely not changed. pCREB data should be supported by histology to determine which cells (TAU positive neurons?) are underlying these changes.

Fig. 1B: Staining showing multiple TAU positive neurons that also show less/no CBP is required. Is the single AT100 positive neuron shown here still alive? If the CBP changes at the neuronal level are such rare events as depicted, then the changes in overall levels (Fig, 1A) may not result from lower expression in neurons.

Fig. 1C: The presented blot for H2B4Kac does not convincingly show level differences. Furthermore, the blot presented here seems not to be from the same series of experiments, since for all other blots n=5 (WT) and n=6 (THY-Tau22) have been loaded, but for the H2B4Kac only 5 samples each were run. This makes comparison and quantification relative to the other blots invalid. This needs to be rectified.

The transcriptome analysis is done at 8 months of age, while the staining and blots are from 12-month-old animals. Why this difference?

Fig. 1D/E: These selected genes need to be confirmed by qPCR to provide validation of RNA-seq data, in an independent cohort. Otherwise the selection is biased.

Fig. 1F: ATF5 is not listed in the figure. Did the authors mean ATF6?

Fig. 2H: Why was Nrp2 chosen as an example? The difference in H2Aac between WT and TAU seems obvious; the difference between the treated TAU groups is not obvious at all. Are the changes in H2Aac of the 477 'overlap' genes between the treated TAU all that small? How meaningful is this and is there really target engagement of CSP-TTK21. In light of the large number of increased genes (Fig. 2A), the authors should present a global method to assess changes in H2Bac, i.e. western blotting. Without such data, target engagement is in question, which raises concerns of the mode of action of CSP-TTK21 in mediating behavioral changes.

Page 10: "...which could be partially restored by CBP HAT activation CSP-TTK21" is an overstatement. Changes in Fig. 2H and Fig EV3A are minimal and not obvious as data is currently presented. Restoration would suggest that H2Bac levels become comparable to WT.

Fig. 3-5: This is the most intriguing part of the report. Together with the behavioral changes in Fig. 7, the improvements seem to be mediated by a small number of genes that are differentially regulated between CSP and CSP-TTK1 treated TAU mice (Fig 3A). Without mechanistic validation, however, this part remains a global observation rather than a molecular explanation.

The biggest problem is that there is no real overlap between the genes with changes in H2Bac in CSP-TTK1 treated TAU mice (Fig. 2A) and those changed in memory trained CSP-TTK1 treated TAU mice at the RNA-seq level (Fig. 3A). This raises the question if the behavioral improvements mediated by CSP-TTK1 (Fig. 7) are indeed due to CBP HAT activation or alternative molecular effects.

Page 15: The discussion overstates the effect sizes observed in the present study; the H2B acetylome is not 'normalized', but only marginally changed. The learning-induced transcriptome is not 'restored', but only a small number of genes (like very relevant ones) are changed. The epigenetic signature of TAU mice is not 'corrected'. To make this statement, Fig. 3A needs to compare trained WT mice with CSP-TTK1 treated and trained mice and show that only a very small (or better no) genes are differentially regulated.

This reviewer fully agrees with the authors that the effects of CSP-TTK1 are of therapeutic interest; how these effects are mediated requires further detailed analysis.

Referee #2 (Comments on Novelty/Model System for Author):

See remarks sent to the authors

Referee #2 (Remarks for Author):

In this manuscript the authors looked at the effects of an acetyltransferase activator (CSP-TTK21) in a mouse model with AD-like Tau pathology (THY-Tau22). They performed a genome-wide screening using integrated ChIP-sequencing and RNA-sequencing to determine genes that were altered by CSP-TTK21 in learning conditions, and tested if this HAT activator molecule has an impact on neuronal plasticity and memory in THY-Tau22 mice. They also found that decreased levels of CBP is associated with lower levels of H2B4K in resting condition in THY-Tau22 mice.

-In Fig. 1, the authors showed that the protein level of CBP is decreased in resting condition in THY-Tau22 mice. However, it is necessary to show that CBP level or activity is also decreased following spatial learning to justify the potential usefulness of CSP-TTK21 to prevent plasticity and memory alteration in THY-Tau22 mice.

- Fig 1: The loading control used for Fig. 1A and C should be clearly indicated in the figure legend.

- Fig 3: What is the rationale of using a dose of 500ug/mouse of CSP-TTK21 3 times (1 per week)? The authors did not mention which type of spatial training was performed for 3 days? Is it the MWM and if so, how was it performed during these 3 days. Since the expression level of many genes (such as immediate early genes) is time dependent after learning a behavioral task, it is important to mention when mice were sacrificed after the end of spatial training.

- To validate the efficiency of their treatment, the authors need to show that the CSP-TTK21 treatment used here effectively activated CBP/p300 in the hippocampus following spatial learning in their animal model.

- The number of mice used should be clearly indicated in each figure legends. Is it male or female mice that were used in this study?

- Fig. 7B: A 3-way ANOVA (Day x Genotype x Treatment) instead of a 2-way ANOVA need to be performed. Since escape latencies depend on swimming speed in the MWM, this parameter should be reported. Is the swimming distance to reach the hidden platform the same between groups during the acquisition phase? Fig 7C: A 2-way ANOVA (Genotype x Treatment) instead of a one-way ANOVA need to be performed when comparing groups together, and a 95% confidence interval analysis (not a Student's test) need to be done when comparing groups with the dotted line. During the probe trial, is the time spent in the target quadrant supported by the number of time mice crossed the previous platform location? Fig. 7D: the mean track of each group should be represented. The results of the cued trial also need to be reported.

- p.14: The authors wrote: we observed a deficit in theta burst-induced LTP, which was partly restored with CSP-TTK21 treatment (Fig EV7A). This is not accurate since no significant difference was observed between THY-Tau22 CSP and THY-Tau22 CSP-TTK21 treated mice.

- The authors should discuss why CSP-TTK21 increased acetylation of H2B in H3K27ac enriched super-enhancer-regulated genes.

- p.12 The authors should discuss why up-regulated genes in CSP-TTK21 vs CSP-treated THY-Tau22 mice were not significantly correlated with changes in H2Bac levels.

Referee #3 (Remarks for Author):

The manuscript by Chatterjee and colleagues investigates the beneficial effects of CSP-TTK21, a small molecule that enhances CBP/p300 histone acetyltransferase activity, in a mouse model of tauopathy. The authors demonstrate that synaptic plasticity and memory deficiencies can be restored along with histone H2B acetylation and specific transcriptional defects. This is a thorough study that presents some novel and relevant findings. Previous studies focused on HDAC inhibitors rather than HAT activators, therefore, the study is potentially interesting and relatively novel. However, the general structure of the manuscript should be improved, and I do not agree with the way most of the bioinformatical analyses were conducted. I also have concerns regarding the reproducibility of the results given the apparent differences across replicates in ChIP-seq experiments.

Main concerns:

1. The abstract and the first paragraph of Discussion introduce the study indicating first that CSP-TTK21 corrects some deficits in a mouse model of tauopathy, and summarizing later the investigation of the genomic events that may underlie this correction. However, surprisingly, in the Results section, the authors describe first the analysis of ChIP-seq and RNA-seq data and only in the last section of Results present the impact of CSP-TTK21 in behavior and physiology (which would indeed provide the rationale for the study). I believe that the manuscript should have the same logical flow outlined in the abstract.
2. Figure EV2 shows that the overlap between replicates is very low (between 10-20% of the differential H2Bac peaks). This creates serious concerns regarding the subsequent bioinformatic analyses. The authors should present PCA graphs and Pearson correlation heatmaps (for example) to demonstrate the similarity of their ChIP-seq and RNA-seq replicates and support the robustness of their analyses and the reliability of their conclusions.
3. Most of the bioinformatical analyses seem amateurish and lack statistical rigor. For example, the heatmaps presented in Figure 4, 5 and EV5 are misleading and confusing.
4. The most serious criticisms regarding bioinformatics analyses refer to the abusive use of Venn diagrams. This type of graph can be a useful representation of similarities and divergences, but bears little quantitative and statistical information (they are based on arbitrary thresholds and ignore quantitative information). For example, to define the set of H2B hypoacetylated regions in THY-Tau22 mice, rather than rely all the subsequent analyses in the overlap between two datasets, they could use the information of all the datasets to identify H2Bac-enriched regions through the genome and apply statistical tests to later identify differentially acetylated regions between the two WT and the two THY-Tau22 samples, and between CSP-TTK21 and saline treated mice.
5. The results presented in Figure 3A should illustrate the most important finding in this article: THY-Tau22 and WT mice differ in more than 2500 genes, while CSP-TTK21- and CSP-treated THY-Tau22 only differ in less than 200 genes. However, this partial restoration (less than 10%) seems to be sufficient to ameliorate behavioral, neurological and electrophysiological defects. Unfortunately, the table format presents partial and biased information. Scatter plots, heat maps, line plots, violin plots or other representations (there are many options) should illustrate this recovery providing at the same time quantitative information and statistical parameters.
6. The description of genomic results and Discussion section could be shortened. I would recommend to restrict these sections to the main finding described in the study: the reversal of transcriptional and histone acetylation deficits by CSP-TTK21, eliminating the abundant digressions of the current version of the manuscript (for instance, the discussion of super-enhancers, a controversial entity, seems completely unnecessary). The manuscript might be better suited to publication as a short report in EMBO Molecular Medicine.
7. Similar studies have been conducted with other mouse models of Alzheimer's disease and HDACi rather than HAT activators. It would be very interesting that the article included a direct comparison of the genomic actions of these two families of compounds in these models of disease.
8. Figure 8 is unnecessary. There is a body of literature relating H3K27ac and enhancers. This evidence does not exist for H2Bac therefore the model is highly speculative and based exclusively in the data presented in this study. What are "Identity genes" and where in the manuscript it is shown that they provide "identity"?

Other comments:

9. The results presented in Fig. EV7 are very relevant and should be presented as a main figure.
10. The use of the label "H2B4Kac" is a bit confusing next to other labels such as "H2BK5ac". I would recommend to replace it by H2BKw,x,y,z, in which w-z are replaced by the actual lysine positions investigated (that are not detailed anywhere in the manuscript).
11. Some concepts and abbreviations are used without being introduced (e.g., Tau-Ph404, CSP, the use of inflammatory markers should be supported by appropriate references, etc).
12. The "reduced basal neuronal activity" discussed in page 7 correspond to a reduction of the number of cells activated or to decreased expression of IEG in active cells? The interpretation of either result would be different.
13. How was the "predictive motif analysis" referred in page 7 (Fig. 1F) conducted? There is no description in the text.
14. For comparison, the genomic snapshots could also include the profile of CBP from Kim et al. 2010 (Fig. 2D, Fig EV3).

1st Revision - authors' response

17 July 2018

REVIEWER 1:

The authors show that treatment of TAU transgenic mice with a CBP HAT activator (CSP-TTK1) improves memory deficits and other associated correlates. They also show some changes in histone acetylation and at the transcriptional level. In particular the transcriptional study in mice that have undergone memory training is elegant and may lead to identification of the molecular mode of action of CSP-TTK1 in the future. The study as it stands has a number of experimental shortcomings and at times over-emphasizes outcomes. Furthermore, a direct connection between H2B acetylation and memory improvements in TAU mice is missing. The issues addressed below would need to be addressed in full before a revised version of the manuscript should be re-assessed.

Issues in order of appearance in the manuscript

Fig. 1A: The blots do not convincingly show a reduction in CREB. If CREB levels are indeed reduced, but its phosphorylation not, then relative CREB activity is likely not changed. pCREB data should be supported by histology to determine which cells (TAU positive neurons?) are underlying these changes.

We acknowledge this critical comment because this is the conclusion we also drawn: it is difficult to attest for CREB activity in this series of western blot analyses measuring total CREB and Ser133Phospho-CREB levels. We thus decided to exclude these western blots from the revised version of the manuscript. We had a series of slices from 12 month-old THY-Tau22 mice (that we used for further CBP staining, see below), but post-translational modifications such as phosphorylation are not so stable, so we decided not to undertake such study. However, we think that we now provide enough demonstration highlighting that the cAMP/CBP pathway is significantly altered in THY-Tau22 mice as shown by the RNAseq analysis (Figure 1C), the reduction of CBP levels (Figure 1A,B) and of histone acetylation (Figure EV1A) in tauopathic mice at the age of 12 month, the reduction of cAMP-dependent IEG expression already visible at 8 month of age (Figure 1C).

Fig. 1B: Staining showing multiple TAU positive neurons that also show less/no CBP is required. Is the single AT100 positive neuron shown here still alive? If the CBP changes at the neuronal level are such rare events as depicted, then the changes in overall levels (Fig. 1A) may not result from lower expression in neurons.

We apologize that the immunohistochemical data lead to a misinterpretation of the results. Our aim was to demonstrate that neurons exhibiting aggregated Tau demonstrate altered CBP levels. We now provide new analysis for CBP/Tau co-staining in slices of 12-month old Thy-Tau22 mice (3-4 slices/animal, n=5 mice). We used another CBP antibody reference (Anti-KAT3A / CBP antibody ab2832, Abcam) and found the same results. New pictures performed with an apotome/Zeiss equipment are presented on Figure 1A. These new co-labelings allowed us to observe that indeed, CBP immunoreactivity was decreased in AT100 positive-cells. Reduction of CBP mainly occur in still living cells since only few of AT100 positive cells were DAPI-negative and therefore exhibit ghost tangles. This is now indicated in a representative image showing multiple AT100-positive

neurons with DAPI blue nuclei and one ghost-NFT. AT100-positive cells represent an advanced stage of Tau pathology in these Tau transgenic mice. However, these observations while clearly linking Tau pathology to CBP reduction do not preclude that CBP is reduced in AT100-negative neurons but yet exhibiting conformationally altered Tau species or even in other cell types, likely explaining why we can achieve to demonstrate a CBP reduction by Western Blot.

Fig. 1C: The presented blot for H2B4Kac does not convincingly show level differences. Furthermore, the blot presented here seems not to be from the same series of experiments, since for all other blots n=5 (WT) and n=6 (THY-Tau22) have been loaded, but for the H2B4Kac only 5 samples each were run. This makes comparison and quantification relative to the other blots invalid. This needs to be rectified.

We apologize for this mistake. Western blots of the same series of experiments were performed with n=5-6/group and some with n=5/group. The number of samples is now indicated on the figure and blots are shown separately. In each gel, we have compared samples loaded on a same gel to avoid batch effects.

Regarding quantifications, we have standardized our western blot protocol to minimize heterogeneity, by using PROTEAN TGX Stain-Free™ Precast Gels (4-20%, Biorad). In addition to avoid the differences in gel preparation, we have used stain-free imaging, which allows to normalize the signal detected by the antibody to total proteins. A UV-induced 1-minute reaction of the gels after protein migration produces fluorescence. The fluorophores remain covalently bound to the proteins throughout blotting and may be subsequently visualized in gels or on the nitrocellulose membranes for validation in the western blotting workflow with a ChemiDoc™ imaging Systems apparatus. Thus, to quantify the level of a specific protein in a sample, we normalize it to the level of total protein detected in the gel lane by this method. Modified histones were further normalized to the total histones level. As they are usually run on different gels, the calculation was as follows : the relative quantity of [modified histone/ total proteins] was divided by the relative quantity of [total histone/ total proteins] from the same sample. Usually 5-8ug of protein sample were loaded. All samples were analyzed using this method. A representative blot is shown in Figure S1A. The quantification is shown on the right.

The transcriptome analysis is done at 8 months of age, while the staining and blots are from 12-month-old animals. Why this difference?

The effects of the tauopathy on the CBP-dependent pathway were tested when the tau pathology and memory deficits are maximum in this model and we thus used 12 month-old THY-Tau22 (Schindowski et al., 2006; Van der Jeugd et al., 2013). To lead further studies aimed at testing our therapeutic strategy using HAT activation with CSP-TTK21, we used 8 month-old animals as the pathology is already present, but not maximum (inflammation is still progressing) (Schindowski et al., 2016; Laurent et al., 2017). The transcriptomic studies shown in figure 1C and all subsequent experiments in the paper are performed at this age.

We have added this explanation in the result section of the manuscript (Figure 1) : « In order to test the potential effect of our new drug, we carried out our study at 8 month of age, an earlier [*than 12 month-old*] symptomatic age where mice already show memory deficits and inflammatory processes but the pathology is still progressing (Schindowski et al., 2006). ».

Fig. 1D/E: These selected genes need to be confirmed by qPCR to provide validation of RNA-seq data, in an independent cohort. Otherwise the selection is biased.

As we uncovered that treatment with CSP-TTK particularly impacts on the transcription of genes activated during learning, we considered more appealing to focus our validation experiments in the learning conditions rather than on basal changes. Therefore, we specifically generated a new cohort of 8 month-old mice (n=5 mice/group) to validate these markers using RT-qPCR but also Western Blot. This cohort was raised in parallel to the cohort used for dendritic spine counting (Figure 2B). Four groups of animals were generated : WT VEH_HC, WT VEH, TAU VEH, TAU MOL treated with proper molecules, to match the RNAseq data presented in figure 3. From this independent new cohort, we prepared half of the hippocampi for protein levels quantification (Figure 4G) and the other half for mRNA levels and RT-qPCR validations (Figure EV2B, Figure 4F). Several markers were investigated:

1) We performed RT-qPCR validations of inflammatory response genes as shown in Figure EV2B. As expected, we show a significant induction of GFAP, *Ccl4* and *Itgax* inflammatory markers in THY-Tau22 mice compared to WT mice, which are not significantly affected by the treatment, as mentioned in the result section of the manuscript.

2) We performed RT-qPCR validations of IEGs and other genes: These data are represented in Reviewer 1 Figure 1. Only genes strongly induced by learning or decreased expression by the pathology at the basal level reached significance with one-way ANOVA statistical tests. We validated a significant effect of Tau pathology on the genes: *Mvd* (a gene associated to the cholesterol pathway and found in the top decreased genes in both our RNAseq, Figures 1 and Figure 3 of the revised manuscript), *Kcnh3* (a potassium voltage-gated channel), *Klotho* (a pleiotropic protein that delays aging and enhances cognition ; Kurosu et al., 2005; Dubal et al., 2014) and *Neurotensin* (a peptide recently shown to exert functions in learning and memory; Xiao et al, 2014). We found a significant effect of CSP-TTK21 treatment on the Neurotensin gene. Learning significantly induced *c-Fos*, *Egr-1* and *Klotho* gene expression. We may emphasize here that the RT-qPCR technique is less sensitive than RNAseq, which makes it difficult to address low fold change validations.

3) We had led in parallel, a study aimed at identifying the effect of long-term administration of CSP-TTK21 to the THY-Tau22 mice colony. Thus we performed a chronic treatment from 3 month to 8 month of age and tested the mice in the Morris water maze before euthanasia. As the probe test was performed 10 days after training, and the mice were killed 48 hours after the probe test, they could be considered as resting mice. In this experiments, we have tested the ability of CSP-TTK21 to restore gene expression of IEGs in basal conditions. The result is presented in Reviewer 1 Figure 2. We did not include these data in the present manuscript as, with the other experiments performed, it will be a matter of another publication. Nevertheless, we confirmed the down-regulation of several IEGs (*Arc*, *Egr-1* and *Nr4a1*) at basal level and their induction after CSP-TTK21 treatment (see Reviewer 1 Figure 2).

Overall, we have validated several genes by RT-qPCR in different two cohorts of animals, investigating basal and learning conditions. Importantly, we tested the protein expression of two important genes that could have had a direct impact on the cognitive functions of THY-Tau22 mice : *Klotho* and *Neurotensin*. We interestingly observed that these proteins were both decreased in the pathology (TAU VEH vs WT VEH) and increased by the CSP-TTK21 molecule (TAU VEH vs TAU MOL) supporting their role in the beneficial effect of our molecule. This is now discussed in the revised version of the manuscript.

Fig. 1F: ATF5 is not listed in the figure. Did the authors mean ATF6?

Yes, we meant ATF6. This is now corrected.

Fig. 2H: Why was *Nrp2* chosen as an example?

Nrp2 was chosen as a typical example of a gene, which gene body was differentially enriched in H2Bac upon the different experimental conditions. *Nrp2* displays neuronal functions with demonstrated implication in the formation of the neuronal circuits in the hippocampus of the adult. This is a representative example that illustrates the profile of this class of genes. Additional genes such as *Epha4* or *Gria1* were also chosen and shown in a supplemental figure. This class of genes presenting a broad profile enrichment in H2Bac (and H3K27ac) is now represented in supplemental figure 7B,C.

The difference in H2Aac (H2Bac) between WT and TAU seems obvious; the difference between the treated TAU groups is not obvious at all. Are the changes in H2Aac (H2Bac) of the 477 'overlap' genes between the treated TAU all that small?

Global changes of this class of genes, which are highly expressed, are shown in Figure EV4B,C. Genes like *Nrp2*, regulated by super-enhancers, are enriched in the 477 overlapping genes, but they indeed do not represent the genes that carry the most differences in terms of fold changes upon treatment with the molecule. However, they are significant and replicated. We originally focused on this class of genes because we found a correlation with the fact that these genes were down-regulated upon learning and upon treatment by the molecule. However, we could not draw solid conclusions in terms of mechanistics, and this would indeed require more analyses. We have

focused in the revised version of the manuscript on more obvious changes, that occur at TSS and CBP enhancers (see Figures 7A-D and EV4A).

How meaningful is this and is there really target engagement of CSP-TTK21. In light of the large number of increased genes (Fig. 2A), the authors should present a global method to assess changes in H2Bac, i.e. western blotting. Without such data, target engagement is in question, which raises concerns of the mode of action of CSP-TTK21 in mediating behavioral changes.

We have assessed H2Bac by western blot analyses, because it is true that H2Bac seems widely increased after CSP-TTK21 treatment (Figure 6D, 6H). At this age (8 month-old mice), we cannot detect any change in H2Bac between WT and THY-Tau22 mice as assessed at the global level (i.e. WB analyses). Nevertheless, we have compared acetylation levels in the batch of animals that received 3 CSP-TTK21 injections and were tested in the Morris water maze (Figure 1D). Animals were killed 4 days after the probe test. We measured a significant increase of H2BK5K10K15K20 and H3K27ac, and an increased tendency for H2BK5ac. We present these data in Figure EV1I. It remains that ChIP-seq is a puissant technique to measure genomewide histone acetylation changes at specific loci, and that we clearly evidenced significant differential changes for H2Bac in THY-Tau22 mice and also in response to the treatment (Figures 6 and 7), which is more powerful and precise than western blots.

Page 10: "...which could be partially restored by CBP HAT activation CSP-TTK21" is an overstatement. Changes in Fig. 2H and Fig EV3A are minimal and not obvious as data is currently presented. Restoration would suggest that H2Bac levels become comparable to WT.

We agree that the restoration is moderate on these genomic loci in terms of fold change (Figure EV4B). As discussed above, we focused the revised manuscript on the larger changes observed at TSS and CBP enhancers. We thank the reviewer for the suggestion to compare H2Bac levels reached in the TAU MOL condition vs. the WT VEH to measure the rescue. We did this comparison which is presented in figure 6H (all loci) and EV3E (increased and decreased loci enrichment separately). Although the rescue was not complete with respect to fold change, we found that 95% of the deregulated loci showed a partial H2B re-acetylation (Figure 6E) in both replicates (Figure EV3D) after CSP-TTK21 treatment.

Fig. 3-5: This is the most intriguing part of the report. Together with the behavioral changes in Fig. 7, the improvements seem to be mediated by a small number of genes that are differentially regulated between CSP and CSP-TTK1 treated TAU mice (Fig 3A). Without mechanistic validation, however, this part remains a global observation rather than a molecular explanation.

CSP-TTK21 indeed modifies only a subset of genes relative to all the number of deregulated genes found in CSP-treated mice ($\approx 10\%$). Here again, we checked the rescued learning transcriptome by comparing remaining differential genes in the TAU MOL vs WT VEH conditions. We found it was rescued by about 50% (Figure 3D-F), which could account to improving the phenotype of Thy-Tau22 mice. Mechanistically, this tells us that the beneficial effects of CSP-TTK21 are probably both direct and indirect. This is now discussed in the revised manuscript. We went further into these mechanisms and found that CSP-TTK21 activated targets could be Klotho and Neurotensin, for which we validated an increase in protein expression (Figure 4). There is a growing literature around Klotho, which is documented as exerting beneficial effects in aging, and more recently in a mouse model of AD (Kurosu et al., 2005 ; Leon et al., 2017; Masso et al., 2017; Kuang et al., 2017). The neurotensin system was also found altered in the temporal lobe of AD patients (Gahete et al., 2010) and microinjections of Neurotensin have been shown to restore spatial memory functions and neuronal excitability in APP/PS1 mice (Xiao et al, 2014). Thus, we think that by defining (at least) these specific targets, we have improved the understanding of the mechanisms likely involved in beneficial effect of CSP-TTK21 in THY-Tau22 mice.

The biggest problem is that there is no real overlap between the genes with changes in H2Bac in CSP-TTK1 treated TAU mice (Fig. 2A) and those changed in memory trained CSP-TTK1 treated TAU mice at the RNA-seq level (Fig. 3A). This raises the question if the behavioral improvements mediated by CSP-TTK1 (Fig. 7) are indeed due to CBP HAT activation or alternative molecular effects.

In our study, we have come to the conclusion that there was no correlation between increased/decreased gene expression (either in resting or in behaving mice) and H2Bac enrichment in resting mice. This holds true for Klotho and Neurotensin as well. The Neurotensin locus is actually very poorly acetylated with H3K27ac/H2Bac. So how CSP-TTK21 actually leads to increased transcription of these genes is still unresolved but may not solely rely on H2Bac levels. It may also involve other histone marks, or increased CBP/p300 co-activator efficiency. A lack of clear correlation between acetylation levels at specific genes and their transcriptional status, either in learning and memory (Lopez-Atalaya and Barco, 2014; Halder et al., 2016) or in response to HDAC inhibition (Lopez-Atalaya et al., 2013) has been reported previously. Other epigenetic mechanisms such as DNA methylation were found to more readily reflect the transcriptome status (Halder et al., 2016). As suggested in the discussion section, a possibility is that the CBP/p300 - H2Bac balance may have a role in the tridimensional organization of the neuronal chromatin (Mitchell et al., 2014 ; Watson and Tsai, 2017) which might favor a correct spatiotemporal gene expression program in response to learning.

Page 15: The discussion overstates the effect sizes observed in the present study; the H2B acetylation is not 'normalized', but only marginally changed. The learning-induced transcriptome is not 'restored', but only a small number of genes (like very relevant ones) are changed. The epigenetic signature of TAU mice is not 'corrected'. To make this statement, Fig. 3A needs to compare trained WT mice with CSP-TTK1 treated and trained mice and show that only a very small (or better no) genes are differentially regulated.

According to this comment, we have moderated the messages in the discussion section of the revised version of the manuscript. However, the H2Bac rescue as calculated according to the reviewer's suggestion, is of 95%. It is therefore quite significant (even if not in terms of fold change, see discussion above). In addition, there were only a few loci that still showed deregulation (Figure 6H and Figure EV3E).

This reviewer fully agrees with the authors that the effects of CSP-TTK1 are of therapeutic interest; how these effects are mediated requires further detailed analysis.

We thank the reviewer for this encouraging remark. We hope that the revised version of the manuscript, presenting new analyses of the previous data, new data on specific regulators involved in learning and memory, as well as focusing on more obvious changes detected in response to the molecule, will convince the reviewer that the paper is of interest to the readership of *EMBO Molecular Medicine*.

References

- Gahete MD, Rubio A, Córdoba-Chacón J, Gracia-Navarro F, Kineman RD, Avila J, Luque RM, Castaño JP. Expression of the ghrelin and neurotensin systems is altered in the temporal lobe of Alzheimer's disease patients. *J Alzheimers Dis.* 2010;22(3):819-28.
- Halder R, Hennion M, Vidal RO, Shomroni O, Rahman RU, Rajput A, Centeno TP, van Bebber F, Capece V, Garcia Vizcaino JC, Schuetz AL, Burkhardt S, Benito E, Navarro Sala M, Javan SB, Haass C, Schmid B, Fischer A, Bonn S. DNA methylation changes in plasticity genes accompany the formation and maintenance of memory. *Nat Neurosci.* 2016 Jan;19(1):102-10.
- Kuang X, Zhou HJ, Thorne AH, Chen XN, Li LJ, Du JR. Neuroprotective Effect of Ligustilide through Induction of α -Secretase Processing of Both APP and Klotho in a Mouse Model of Alzheimer's Disease. *Front Aging Neurosci.* 2017 Nov 2;9:353.
- Kurosu H, Yamamoto M, Clark JD, Pastor JV, Nandi A, Gurnani P, McGuinness OP, Chikuda H, Yamaguchi M, Kawaguchi H, Shimomura I, Takayama Y, Herz J, Kahn CR, Rosenblatt KP, Kuro-o M. Suppression of aging in mice by the hormone Klotho. *Science.* 2005 Sep 16;309(5742):1829-33.
- Laurent C, Dorothée G, Hunot S, Martin E, Monnet Y, Duchamp M, Dong Y, Légeron FP, Leboucher A, Burnouf S, Faivre E, Carvalho K, Caillierez R, Zommer N, Demeyer D, Jouy N, Sazdovitch V, Schraen-Maschke S, Delarasse C, Buée L, Blum D. Hippocampal T cell infiltration promotes neuroinflammation and cognitive decline in a mouse model of tauopathy. *Brain.* 2017 Jan;140(1):184-200.

Leon J, Moreno AJ, Garay BI, Chalkley RJ, Burlingame AL, Wang D, Dubal DB. Peripheral Elevation of a Klotho Fragment Enhances Brain Function and Resilience in Young, Aging, and α -Synuclein Transgenic Mice. *Cell Rep*. 2017 Aug 8;20(6):1360-1371.

Lopez-Atalaya JP, Barco A. Can changes in histone acetylation contribute to memory formation? *Trends Genet*. 2014 Dec;30(12):529-39.

Lopez-Atalaya JP, Ito S, Valor LM, Benito E, Barco A. Genomic targets, and histone acetylation and gene expression profiling of neural HDAC inhibition. *Nucleic Acids Res*. 2013 Sep;41(17):8072-84.

Massó A, Sánchez A, Bosch A, Giménez-Llort L, Chillón M. Secreted α Klotho isoform protects against age-dependent memory deficits. *Mol Psychiatry*. 2017 Oct 31.

Mitchell AC, Javidfar B, Bicks LK, Neve R, Garbett K, Lander SS, Mirnics K, Morishita H, Wood MA, Jiang Y, Gaisler-Salomon I, Akbarian S. Longitudinal assessment of neuronal 3D genomes in mouse prefrontal cortex. *Nat Commun*. 2016 Sep 6;7:12743.

Schindowski K, Bretteville A, Leroy K, Bégard S, Brion JP, Hamdane M, Buée L. Alzheimer's disease-like tau neuropathology leads to memory deficits and loss of functional synapses in a novel mutated tau transgenic mouse without any motor deficits. *Am J Pathol*. 2006 Aug;169(2):599-616.

Van der Jeugd A, Vermaercke B, Derisbourg M, Lo AC, Hamdane M, Blum D, Buée L, D'Hooge R. Progressive age-related cognitive decline in tau mice. *J Alzheimers Dis*. 2013;37(4):777-88.

Watson LA, Tsai LH. In the loop: how chromatin topology links genome structure to function in mechanisms underlying learning and memory. *Curr Opin Neurobiol*. 2017 Apr;43:48-55.

Xiao Z, Cilz NI, Kurada L, Hu B, Yang C, Wada E, Combs CK, Porter JE, Lesage F, Lei S. Activation of neurotensin receptor 1 facilitates neuronal excitability and spatial learning and memory in the entorhinal cortex: beneficial actions in an Alzheimer's disease model. *J Neurosci*. 2014 May 14;34(20):7027-42.

Figure 1: RT-qPCR validations of RNA-seq data.:

A Time line of the experiment and experimental groups: 8 month-old mice were injected 3 times (1 per week) with Vehicle (WT mice, WT VEH, saline), Vehicle (THY-Tau22 mice (TAU VEH), CSP 500 µg /mouse) or Molecule (THY-Tau22 mice (TAU MOL), CSP-TTK21 500 µg/mice). One subgroup of WT mice was left in their Home cage (WT VEH_HC) and the other mice (WT and THY-Tau22) were subjected to 3 days of spatial training (Learning). RNA extracts were isolated from the dorsal hippocampus, one hour after the last training (n=5/group).

B RNA-seq data result from a first cohort of mice (n=5/group). Statistical analysis was performed with the method proposed by Love et al. (2014) (Love et al, 2014) implemented in the DESeq2 Bioconductor library (v1.0.19). *Mvd*: \$ pathology p=1,17E-27 for WT VEH vs. TAU VEH; *Cyp46a1*: \$ pathology p=1,80E-13 for WT VEH vs. TAU VEH, * learning p=0,0168 for WT VEH vs. WT VEH_HC, # molecule p=4.00E-05 for TAU MOL vs. TAU VEH; *Kcnh3*: \$ pathology p=1.06E-27 for WT VEH vs. TAU VEH, # molecule p=0.0054 for TAU MOL vs. TAU VEH; *cfos*: \$ pathology p=0.0276 for WT VEH vs. TAU VEH, * learning p=2.5E-27 for WT VEH vs. WT VEH_HC, # molecule p=0.0016 for TAU MOL vs. TAU VEH; *Egr-1*: \$ pathology p=1.66E-06 for WT VEH vs. TAU VEH, * learning p=8.69E-12 for WT VEH vs. WT VEH_HC, # molecule p=1.19E-06 for TAU MOL vs. TAU VEH; *Kl*: * learning p=2.78E-22 for WT VEH vs. WT VEH_HC, \$ pathology p=1.73E-32 for WT VEH vs. TAU VEH, # molecule p=1.73E-14 for TAU MOL vs. TAU VEH; *Nts*: \$ pathology p=0.0110 for WT VEH vs. TAU VEH, # molecule p=7.34E-06 for TAU MOL vs. TAU VEH.

C RT-qPCR result from another cohort of mice (n=5/group). One-way ANOVA with uncorrected Fisher's test. *Mvd*: F(3,16)=31.69, p<0.0001; \$ pathology p=0.0001 for WT VEH vs. TAU VEH; *Cyp46a1*: F(3,14)=1.777, p=0.1977; *Kcnh3*: F(3,16)=2.469, p=0.0993; \$ pathology p=0.0222 for WT VEH vs. TAU VEH, (*) learning indicates a tendency p=0.0695 for WT VEH vs. WT VEH_HC; *cfos*: F(3,16)=12.08, p=0.0002; * learning p=0.0002 for WT VEH vs. WT VEH_HC; *Egr-1*: F(3,16)=15.99, p<0.0001; * learning p=0.0001 for WT VEH vs. WT VEH_HC. *Kl*: F(3,16)=2,949, p=0.0643; * learning p=0.0145 for WT VEH vs. WT VEH_HC, \$ pathology p=0.0395 for WT VEH vs. TAU VEH. *Nts*: F(3,14)=4,290, p=0.0241; \$ pathology p=0.0036 for WT VEH vs. TAU VEH, # molecule p=0.0081 for TAU MOL vs. TAU VEH.

Figure 2: RNA-seq data of figure 1 and RT-qPCR analyses of chronic treatment in basal conditions.

A RNA-seq data result from a first cohort of mice ($n=3-2/\text{group}$). Statistical analysis was performed with the method proposed by Love et al. (2014) (Love et al, 2014) implemented in the DESeq2 Bioconductor library (v1.0.19). * when WT VEH is significantly different from TAU VEH. *Arc*: $p=8,12E-15$; *c-Fos*: $p=1,95E-08$; *Egr-1*: $p=2,57E-12$; *Dusp1*: $p=8,35E-09$; *Nr4a1*: $p=5,26E-56$.

B Time line of the experiment with chronic injections in THY-Tau22 mice: 8 month-old mice were injected 10 times (1 per 2 weeks) with Vehicle (WT mice, WT VEH, saline), Vehicle (THY-Tau22 mice (TAU VEH), CSP 500 μg /mouse) or Molecule (THY-Tau22 mice (TAU MOL), CSP-TTK21 500 μg /mice), starting at the age of 3 months. Mice were tested for spatial memory in the Morris water maze 5 days after the last injection, probe test was performed 10 days after and mice were killed 48hr after. RNA extracts were isolated from the dorsal hippocampus ($n=6/\text{group}$).

C RT-qPCR data resulting from the cohort of mice in B ($n=6/\text{group}$). One-way ANOVA with uncorrected Fisher's test. *Arc*: $F(2,15)=12.55$, $p=0,0006$; \$ pathology $p=0,0002$ for WT VEH vs. TAU VEH, # molecule $p=0,0047$ for TAU MOL vs. TAU VEH; *c-Fos*: $F(2,15)=1.323$, $p=0,2957$, ns; *Egr-1*: $F(2,15)=4.349$, $p=0,0324$; \$ pathology $p=0,0487$ for WT VEH vs. TAU VEH, # molecule $p=0,0128$ for TAU MOL vs. TAU VEH; *Dusp1*: $F(2,15)=0.2095$, $p=0,8133$; ns; *Nr4a1*: $F(2,15)=3.925$, $p=0,0426$; \$ pathology $p=0,0297$ for WT VEH vs. TAU VEH, # molecule $p=0,0270$ for TAU MOL vs. TAU VEH.

REVIEWER2:

In this manuscript the authors looked at the effects of an acetyltransferase activator (CSP-TTK21) in a mouse model with AD-like Tau pathology (THY-Tau22). They performed a genome-wide screening using integrated ChIP-sequencing and RNA-sequencing to determine genes that were altered by CSP-TTK21 in learning conditions, and tested if this HAT activator molecule has an impact on neuronal plasticity and memory in THY-Tau22 mice. They also found that decreased levels of CBP is associated with lower levels of H2B4K in resting condition in THY-Tau22 mice.

-In Fig. 1, the authors showed that the protein level of CBP is decreased in resting condition in THY-Tau22 mice. However, it is necessary to show that CBP level or activity is also decreased following spatial learning to justify the potential usefulness of CSP-TTK21 to prevent plasticity and memory alteration in THY-Tau22 mice.

We found that CBP protein levels were significantly decreased in THY-Tau22 mice of 12 month of age, but CBP levels were not changed in 8 month-old mice at the global level, at least, it was not detectable by western blot analyses. We did not try to assess CBP activity because this would require to perform an immunoprecipitation followed by an HAT activity measurement (a global measurement would result in a confounding effect taken into account the multiple different HATs). One has to acknowledge that CBP IPs are very difficult due to the lack of good IP antibodies, and especially because we are working in heterogenous tissues (vs cell lines or cell cultures). In addition, another level of difficulty resides in the fact that we would have to test WT versus THY-Tau22 and basal vs behaving mice, thereby increasing the number of animals to be tested.

However, we do not think it invalidates our strategy, which is justified by the fact that we show a decrease of H2B acetylation at many genomic loci by ChIP-sequencing (at 8 month of age) and further at the global level by western blot analyses (at 12 month of age). H2Bac is one of the preferred histone modification targeted by CBP in the hippocampus (Alarcon et al., 2004; Valor et al., 2011). In addition, some beneficial effects (i.e. amelioration of learning and memory) have been reported after lentiviral CBP gene transfer in the hippocampus of a mouse model of Alzheimer's disease (Caccamo et al., 2010). The use of an acetyltransferase activator targeting CBP/p300 HAT function can thus be justified to re-establish H2B acetylation levels, whatever the mechanisms underlying this loss.

Lastly, our ChIP-sequencing results indirectly show that CBP acetylation activity is likely reduced in THY-Tau22 mice as H2Bac levels were decreased specifically at CBP enhancers in TAU VEH vs WT VEH mice, when these levels were increased in TAU MOL vs TAU VEH mice (see Figure 7A,B in the revised version of the manuscript).

Of note, a direct evidence of HDAC dysfunction is barely never tackled when HDAC inhibitors are tested in AD or other neurodegenerative diseases models. In other words, many studies except a few (Graff et al., 2012 ; Benito et al., 2015) have reported the effect of HDAC inhibitors in one or the other disease, without testing HDAC levels. Lastly, we would like to emphasize that ChIP-seq analyses have been developed to indeed precising levels at specific genomic loci and thus stands an valid demonstration that CBP activity is increased by CSP-TTK21 treatment, as it increased H2Bac at CBP enhancers (Figure 7A,B).

- Fig 1: The loading control used for Fig. 1A and C should be clearly indicated in the figure legend.

We are using specific gels purchased at Biorad, that allow the detection of the total proteins present in the gels after a short UV illumination. We have used the total amount of proteins present in each lane after transfer onto the nitrocellulose membrane to normalize for protein quantity loading. In addition, on each series of blots, several loading controls (e.g. actin) are tested to ensure homogeneity. Different amounts (1:2, 1:1 and 2:1) of proteins are also loaded on the gels to ascertain for the semi-quantitative aspect.

We agree this was not clearly mentioned in the first version of the manuscript and a new paragraph has been added in the material and method section :

« Proteins were loaded on Midi-PROTEAN TGX Stain-Free™ Precast Gels (26 wells, 4-20%, Biorad). The use of stain-free imaging allows for the normalization of bands to the total protein on a blot, eliminating the use of housekeeping proteins. A UV-induced 1-minute reaction of the gels after

protein migration produce fluorescence. The fluorophores remain covalently bound to the proteins throughout blotting and may be subsequently visualized in gels or on the nitrocellulose membranes for validation in the western blotting workflow. ». This is also mentioned in the figure legends : « Normalization was performed on the total amount of proteins transferred onto the membranes ».

However, when testing histone modifications, the amount of modified histones are normalized to the amount of total histone, as this latter could differ between experimental conditions. So we calculate the amount of modified histone on the amount of total proteins within the corresponding nitrocellulose membrane for each lane, normalized to the amount of total histone on the amount of total proteins within the corresponding nitrocellulose membrane. This is specified as « Acetylated histone/ total histone » in the figures (Fig. S1A or S1I).

- Fig 3: What is the rational of using a dose of 500ug/mouse of CSP-TTK21 3 times (1 per week)?

The rational to use 500 µg/mouse of CSP-TTK21 which corresponds to about 20mg/kg of body weight, comes from our previous study performed with this molecule in WT mice (Chatterjee et al., 2013). In this study, we provided evidence that one injection of CSP-TTK21 efficiently acetylated brain chromatin, favors maturation and differentiation of adult neuronal progenitors in the hippocampus and extended spatial memory duration.

The 3-time injection protocol used here in THY-Tau22 mice comes from the fact that we tried one experiment with only one dose and we did not measure any memory improvement, likely because these are pathological mice, and symptoms can not be recovered following one single shot. Therefore, we decided to increase the number of injections and tested a 3-time injection protocol.

The frequency of injection (1 per week) comes also from our previous study (Chatterjee et al., 2013) showing that the molecule reaches a maximum of presence in the brain within 3 days and goes back to low levels within 7 days (see Figure 3). So we thought that by injecting 1 time per week, we could induce 3 « waves » of increased acetylation in the brain and that may be sufficient to improve the phenotype of these mice.

The authors did not mention which type of spatial training was performed for 3 days? Is it the MWM and if so, how was it performed during these 3 days. Since the expression level of many genes (such as immediate early genes) is time dependent after learning a behavioral task, it is important to mention when mice were sacrificed after the end of spatial training.

The training is for spatial reference memory in the Morris Water Maze (4 trails/days, fixed position of the platform). The reviewer is right, that IEGs are rapidly increased after learning, that is why mice were killed 1hr after the last training on the 3rd day, while other late genes could be regulated by the 3-day training during the process of memory formation. The rational of this protocol can be found on our previous studies (Bousiges et al., 2010), as 3 days of training represent a sufficiently long time-period to form, while still consolidating memory.

We apologize for this oversight. This has been clarified in the figure legend and in the method sections.

- To validate the efficiency of their treatment, the authors need to show that the CSP-TTK21 treatment used here effectively activated CBP/p300 in the hippocampus following spatial learning in their animal model.

As already discussed in the first question from this reviewer, this is technically challenging to evaluate HAT activity after immunoprecipitation from the dorsal hippocampus because anti-CBP antibodies are difficult to work with, especially in tissues. Nevertheless, we think that we bring a clear answer to this question with the figure (now numbered) 7A,B. Indeed, it shows that acetylation of H2B is increased on CBP enhancers in the dorsal hippocampus of TAU mice following 3 injections of CSP-TTK21. We think this is convincing evidence that CSP-TTK21 affects acetyltransferase activity – likely CBP - in this tissue.

- The number of mice used should be clearly indicated in each figure legends. Is it male or female mice that were used in this study?

The number of mice used in each study is now clearly identified in each figure legend, as it is also a request of the journal policy.

We used only males in the study (all behavior experiments, RNA-seq, RT-qPCR, dendritic spines, LTD/LTP, Western blots, immunohistochemistry) except for the ChIP-seq study in which replicates were performed with males and females.

- Fig. 7B: A 3-way ANOVA (Day x Genotype x Treatment) instead of a 2-way ANOVA need to be performed. Since escape latencies depend on swimming speed in the MWM, this parameter should be reported. Is the swimming distance to reach the hidden platform the same between groups during the acquisition phase? Fig 7C: A 2-way ANOVA (Genotype x Treatment) instead of a one-way ANOVA need to be performed when comparing groups together, and a 95% confidence interval analysis (not a Student's test) need to be done when comparing groups with the dotted line. During the probe trial, is the time spent in the target quadrant supported by the number of time mice crossed the previous platform location? Fig. 7D: the mean track of each group should be represented. The results of the cued trial also need to be reported.

Fig. 7B: A 3-way ANOVA (Day x Genotype x Treatment) instead of a 2-way ANOVA need to be performed.

In our experimental protocol, it is not possible to apply a 3-way ANOVA (Day x Genotype x Treatment), because CSP/CSP-TTK21 molecule were not tested in the WT group. Thus we want to compare WT Vehicle (saline solution) to TAU Vehicle (CSP) and Molecule (CSP-TTK21). To do so, we can only perform a 2-way ANOVA (Groups further defined as: WT VEH, TAU VEH and TAU MOL) followed by a multiple comparisons test (Newman Keuls).

An additional observation to explain our statistical choice is based on the fact that the THY-Tau22 mice treated with CSP-TTK21 (TAU MOL) and the WT saline (WT VEH) groups behave the same way. We decided to focus our analyses on treatment effect, knowing that the post-hoc analysis allowed us to focus more precisely on potential differences that could be linked to the genotype (WT VEH vs. TAU VEH). For example, there is no global effect of genotype in probe trial, most probably because the TAU MOL mice behave as the WT VEH mice. A group by group comparison is thus necessary to study the impact of CSP-TTK21 on memory recall in the THY-Tau22 strain.

Since escape latencies depend on swimming speed in the MWM, this parameter should be reported. Is the swimming distance to reach the hidden platform the same between groups during the acquisition phase?

We first decided to show the escape latencies because the swimming speed did not differ depending on the day ($F(4,148)=0.95$, $p=0.46$) or on the treatment ($F(2,37)=1.06$, $p=0.35$). The distance travelled to reach the hidden platform is now added as Supplemental figure (Figure EV1D) as well as the average swimming speed (Figure EV1E).

Fig 7C: A 2-way ANOVA (Genotype x Treatment) instead of a one-way ANOVA need to be performed when comparing groups together, and a 95% confidence interval analysis (not a Student's test) need to be done when comparing groups with the dotted line.

As for the previous comment, our experimental protocol does not allow to perform a 2-way ANOVA (Genotype x Treatment), and this is why we performed a one-way ANOVA followed by multiple comparisons.

We decided to use the Student's test - with a confidence interval fixed at 95% - for two reasons.

First, Student t-test aimed to compare the mean time spent in the target quadrant to a constant value, i.e. the random level in each group (15s). To our knowledge, the ANOVA analysis does not allow that (notice that t values are the square root of F values). Second, as the TAU MOL behave as the WT VEH, the global effect of genotype in probe trial would have masked the treatment effect. This is not due to a lack of treatment efficiency but to the experimental strategy that we chose to apply (i.e. treating the WT mice with saline). Focusing our analyses on treatment effect and using post-hoc comparisons where relevant, make it possible to look more precisely at several groups and several comparisons (genotype: WT VEH vs. TAU VEH, treatment: TAU VEH vs. TAU MOL), the most important of which being between TAU VEH and TAU MOL.

Perhaps can the reviewer consider that we would have had more power with a 2-way ANOVA. Despite its weaker power, the one-way ANOVA followed by multiple comparisons generated significant differences. Furthermore, the reviewer could be concerned by the fact that our WT mice were not treated with CSP but with saline. Therefore part of the deficits found in the TAU VEH mice given CSP could be due to CSP. In fact, we have experimental evidence showing that CSP does not affect behavior in the water maze (see Reviewer 2 Figure). In addition the cognitive status and pathological progression in the THY-Tau22 strain have been largely documented over the last 10 years (e.g. Schindowski et al., 2006; Belarbi et al., 2009; Van der Jeugd et al., 2011; Van der Jeugd et al., 2013; Laurent et al., 2014; Burlot et al., 2015; Ahmed et al., 2015; Laurent et al., 2016; Laurent et al., 2017).

During the probe trial, is the time spent in the target quadrant supported by the number of time mice crossed the previous platform location?

Concerning the number of time that mice crossed the target platform location, we did not found a statistical treatment effect ($F(1,37)=1.89$, $p=0.16$). We performed a student t-test for dependent samples to compare the number of crossings performed into the target platform location and the 3 others (same place in the 3 others quadrants), and highlighted a difference for WT VEH mice ($t(17)=2.62$, $p=0.018$) and TAU MOL mice ($t(13)=2.70$, $p=0.019$), while there is no difference for the TAU VEH mice ($t(10)=0.21$, $p=0.83$). These data have been added in the supplemental figures (Figure EV1F).

We also observed the latency to the first visit to the target platform location compared to the mean latencies to the first visit to 3 others platform locations. We obtained a global treatment effect ($F(1,37)=5.87$, $p=0.008$) that was due to the difference of performances of the TAU VEH mice with the two others groups (significant difference with WT VEH ($p=0.004$) and with TAU MOL ($p=0.012$)). There is no difference between the TAU MOL and WT VEH groups ($p=0.44$). The comparison between the latencies to the first visit to the target platform with the 3 others (same place into the 3 others quadrants), revealed a difference for WT VEH mice ($t(17)=6.90$, $p=0.000004$) and TAU MOL mice ($t(13)=2.30$, $p=0.039$), while there is no difference for the TAU VEH mice ($t(10)=0.82$, $p=0.42$) indicating that the WT VEH and TAU MOL mice went directly toward the target quadrant (TQ) while the TAU VEH mice went indistinctly toward the different other platform locations (3 others). These data have been added in the supplemental figures (Figure EV1G).

Fig. 7D: the mean track of each group should be represented.

We removed the representative tracks and presented instead the tracks corresponding to the “Closest to Mean” and the “Best” performance for each group, which are now presented in the supplemental figures (Figure EV1H).

The results of the cued trial also need to be reported.

Each behavioral test in the Morris water maze, starts with a habituation consisting of a 60s navigation trial in the pool filled with 5cm height of water and a visible platform, as described in the material and method section. The distance from the habituation platform depending on the treatment showed no statistical difference ($F(1,37)=0.17$, $p=0.84$), neither did the percentage of time spent in the four quadrants (no statistical difference: $F(2,37)=0.00$, $p=0.89$). These data have been added in the supplemental figures (Figure EV1C).

- p.14: The authors wrote: we observed a deficit in theta burst-induced LTP, which was partly restored with CSP-TTK21 treatment (Fig EV7A). This is not accurate since no significant difference was observed between THY-Tau22 CSP and THY-Tau22 CSP-TTK21 treated mice.

The reviewer is right, that there was no significance in this experiment. However, the theta-burst induce LTP in TAU MOL animals were pretty much overlapping that in WT VEH animals and separated from that of TAU VEH and we originally thought that it was a good idea to show this tendency. However, because significance was not reached, we decided to discard this information from the manuscript, especially because the main reported deficit from this THY-Tau22 mouse strain is the LTD maintenance, and we had a clear-cut rescue effect on LTD maintenance with the

CSP-TTK21 molecule. So ultimately, the information that CSP-TTK21 promoted a tendency to restore theta-burst induced LTP is dispensable.

- The authors should discuss why CSP-TTK21 increased acetylation of H2B in H3K27ac enriched super-enhancer-regulated genes.

We have detected that H2Bac enriched genomic regions were highly correlated with H3K27ac enriched regions. These data are shown in figures (now numbered) 6, 7 and S7. The percentage of co-localization has also been calculated and is presented in the result section : « *Interestingly, H2Bac peaks were always detected when there was H3K27ac and 43% of H3K27ac- and H2Bac-covered nucleotides co-localized.* ». It is interesting to note that super-enhancers are defined in part by a broad profile H3K27ac covering the gene bodies and these are genomic regions (Hnisz et al., 2013 ; Whyte et al., 2013 ; Achour et al., 2015 ; LeGras et al., 2017) where we found H2Bac enrichment as well. This is an original finding. However, super-enhancers are still not clearly defined in each tissues and our results were too preliminary to be further discussed. This may have led us to over-interpret results in the discussion section as noted by all reviewers.

So our new analyses led us to investigate further on the association of H2Bac at the TSS and on CBP enhancer regions, which we found differentially enriched for H2Bac in our experimental conditions and we kept the focus on these loci rather than on « super-enhancers ».

About the genes that indeed presented increased H2Bac in H3K27ac-enriched super-enhancer regions, they are mentioned in Fig S7B,C with the example of the *neuropilin-2 isoform 1 precursor* gene locus. It has been shown in the literature that the acetylation of H3K27 was also targeted by CBP (Jin et al, 2011; Tie et al, 2014). When aligned together with SeqMiner, H2Bac, H3K27ac and CBP clustered together (Figure 2D of the former version of the manuscript). Therefore, H3K27ac enriched regions are potentially also CBP enriched, that could preferentially induce increased H2Bac levels as well.

- p.12 The authors should discuss why up-regulated genes in CSP-TTK21 vs CSP-treated THY-Tau22 mice were not significantly correlated with changes in H2Bac levels.

This is a very good remark and this question represents now one of the focus in the discussion section.

Text cited p.19-20 of the revised manuscript : « Remarkably, CSP-TTK21 treatment of THY-Tau22 mice resulted in a global increase in H2Bac acetylation at TSS, CBP enhancers (Kim et al., 2010) and some gene bodies .../... A lack of clear correlation between acetylation levels at specific genes and their transcriptional status has been reported previously, either in learning and memory (Lopez-Atalaya and Barco, 2014; Halder et al., 2016) or in response to HDAC inhibition (Lopez-Atalaya et al., 2013). Of note, the specific genomic topologies favor correct spatiotemporal gene expression programs in response to challenge (Mitchell et al., 2014 ; Madabhushi et al., 2015 ; Watson and Tsai, 2017) and the CBP/p300-H2Bac balance may be an important player in such mechanisms.»

Of note, H2Bac levels on genomic loci were measured by ChIP-seq studies in the different experimental conditions WT VEH, TAU VEH and TAU MOL, but in resting animals while this question tackles its comparison to transcriptomics obtained in behaving animals. So we can not exclude that H2Bac enrichments would be different in behaving animals and thus, we can not speculate too much about why this is not correlative.

A Acquisition**B 24hr- retention test****C 10 day- retention test****Effect of CSP (Vehicle) injection on spatial memory performances in WT and THY-Tau22 mice.**

A WT and THY-Tau22 mice ($n=8/\text{group}$) were injected 3 times (1 per week) with CSP ($500 \mu\text{g}/\text{mouse}$) before training of spatial memory in the Morris water maze (MWM); retention (Probe test) was tested on day 4 in an intermediate probe test (24hr-retention test), and 10 days after the last training session (10 day-retention test). Acquisition (Escape latencies, seconds) and retention performances (Time in target quadrant, seconds) are shown for the 2 groups of mice. Both genotypes displayed significant acquisition of the platform location as a 2-way Anova showed a significant Day effect ($F(4, 28) = 18,16, p < 0,0001$). There was no significant genotype ($F(1,7) = 1,131$) or Day x Genotype ($F(4,28) = 0,08$) effects.

B At the 24hr-retention probe test, there was no significant difference in the time spent in the target quadrant between WT CSP and TAU CSP mice (unpaired t test with equal SD, ns $p=0,1543$), neither in the number of annulus crossings (unpaired t test with equal SD, ns $p=0,1738$).

C At the 10 day-retention test, there was a significant genotype effect in the time spent in the target quadrant (unpaired t test with equal SD, ** $p=0,0021$) and in the number of annulus crossing (unpaired t test with equal SD, * $p=0,0484$).

Bar graphs are mean \pm SEM. TQ, Target Quadrant; O, Other, corresponds to the mean of the 3 other quadrants.

References

- Achour M, Le Gras S, Keime C, Parmentier F, Lejeune FX, Boutillier AL, Neri C, Davidson I, Merienne K (2015) Neuronal identity genes regulated by super-enhancers are preferentially down-regulated in the striatum of Huntington's disease mice. *Human molecular genetics* 24: 3481-3496
- Ahmed T, Blum D, Burnouf S, Demeyer D, Buée-Scherrer V, D'Hooge R, Buée L, Balschun D. Rescue of impaired late-phase long-term depression in a tau transgenic mouse model. *Neurobiol Aging*. 2015 Feb;36(2):730-9.
- Alarcon JM, Malleret G, Touzani K, et al. Chromatin acetylation, memory, and LTP are impaired in CBP+/- mice: a model for the cognitive deficit in Rubinstein-Taybi syndrome and its amelioration. *Neuron* 2004;42:947-959.
- Belarbi K, Schindowski K, Burnouf S, Caillierez R, Grosjean ME, Demeyer D, Hamdane M, Sergeant N, Blum D, Buée L. Early Tau pathology involving the septo-hippocampal pathway in a Tau transgenic model: relevance to Alzheimer's disease. *Curr Alzheimer Res*. 2009 Apr;6(2):152-7.
- Bousiges O, Vasconcelos AP, Neidl R, Cosquer B, Herbeaux K, Panteleeva I, Loeffler JP, Cassel JC, Boutillier AL. Spatial memory consolidation is associated with induction of several lysine-acetyltransferase (histone acetyltransferase) expression levels and H2B/H4 acetylation-dependent transcriptional events in the rat hippocampus. *Neuropsychopharmacology*. 2010 Dec;35(13):2521-37.
- Burlot MA, Braudeau J, Michaelsen-Preusse K, Potier B, Aycirix S, Varin J, Gautier B, Djelti F, Audrain M, Dauphinot L, Fernandez-Gomez FJ, Caillierez R, Laprévotte O, Bièche I, Auzeil N, Potier MC, Dutar P, Korte M, Buée L, Blum D, Cartier N. Cholesterol 24-hydroxylase defect is implicated in memory impairments associated with Alzheimer-like Tau pathology. *Hum Mol Genet*. 2015 Nov 1;24(21):5965-76.
- Caccamo A, Maldonado MA, Bokov AF, Majumder S, Oddo S. CBP gene transfer increases BDNF levels and ameliorates learning and memory deficits in a mouse model of Alzheimer's disease. *Proc Natl Acad Sci U S A* 2010;107:22687-22692.
- Hnisz D, Abraham BJ, Lee TI, Lau A, Saint-Andre V, Sigova AA, Hoke HA, Young RA (2013) Super-enhancers in the control of cell identity and disease. *Cell* 155: 934-947
- Jin Q, Yu LR, Wang L, Zhang Z, Kasper LH, Lee JE, Wang C, Brindle PK, Dent SY, Ge K (2011) Distinct roles of GCN5/PCAF-mediated H3K9ac and CBP/p300-mediated H3K18/27ac in nuclear receptor transactivation. *The EMBO journal* 30: 249-262
- Laurent C, Burnouf S, Ferry B, Batalha VL, Coelho JE, Baqi Y, Malik E, Mariciniak E, Parrot S, Van der Jeugd A, Faivre E, Flaten V, Ledent C, D'Hooge R, Sergeant N, Hamdane M, Humez S, Müller CE, Lopes LV, Buée L, Blum D. A2A adenosine receptor deletion is protective in a mouse model of Tauopathy. *Mol Psychiatry*. 2016 Jan;21(1):97-107
- Laurent C, Dorothée G, Hunot S, Martin E, Monnet Y, Duchamp M, Dong Y, Légeron FP, Leboucher A, Burnouf S, Faivre E, Carvalho K, Caillierez R, Zommer N, Demeyer D, Jouy N, Sazdovitch V, Schraen-Maschke S, Delarasse C, Buée L, Blum D. Hippocampal T cell infiltration promotes neuroinflammation and cognitive decline in a mouse model of tauopathy. *Brain*. 2017 Jan;140(1):184-200.
- Laurent C, Eddarkaoui S, Derisbourg M, Leboucher A, Demeyer D, Carrier S, Schneider M, Hamdane M, Müller CE, Buée L, Blum D. Beneficial effects of caffeine in a transgenic model of Alzheimer's disease-like tau pathology. *Neurobiol Aging*. 2014 Sep;35(9):2079-90.
- Le Gras S, Keime C, Anthony A, Lotz C, De Longprez L, Brouillet E, Cassel JC, Boutillier AL, Merienne K (2017) Altered enhancer transcription underlies Huntington's disease striatal transcriptional signature. *Scientific reports* 7: 42875
- Schindowski K, Bretteville A, Leroy K, Bégard S, Brion JP, Hamdane M, Buée L. Alzheimer's disease-like tau neuropathology leads to memory deficits and loss of functional synapses in a novel mutated tau transgenic mouse without any motor deficits. *Am J Pathol*. 2006 Aug;169(2):599-616.
- Tie F, Banerjee R, Saiakhova AR, Howard B, Monteith KE, Scacheri PC, Cosgrove MS, Harte PJ (2014) Trithorax monomethylates histone H3K4 and interacts directly with CBP to promote H3K27 acetylation and antagonize Polycomb silencing. *Development* 141: 1129-1139
- Valor LM, Pulopulos MM, Jimenez-Minchan M, Olivares R, Lutz B, Barco A. Ablation of CBP in forebrain principal neurons causes modest memory and transcriptional defects and a dramatic reduction of histone acetylation but does not affect cell viability. *J Neurosci* 2011;31:1652-1663.
- Van der Jeugd A, Ahmed T, Burnouf S, Belarbi K, Hamdane M, Grosjean ME, Humez S, Balschun D, Blum D, Buée L, D'Hooge R. Hippocampal tauopathy in tau transgenic mice coincides

with impaired hippocampus-dependent learning and memory, and attenuated late-phase long-term depression of synaptic transmission. *Neurobiol Learn Mem.* 2011 Mar;95(3):296-304.

Van der Jeugd A, Vermaercke B, Derisbourg M, Lo AC, Hamdane M, Blum D, Buée L, D'Hooge R. Progressive age-related cognitive decline in tau mice. *J Alzheimers Dis.* 2013;37(4):777-88.

Whyte WA, Orlando DA, Hnisz D, Abraham BJ, Lin CY, Kagey MH, Rahl PB, Lee TI, Young RA (2013) Master transcription factors and mediator establish super-enhancers at key cell identity genes. *Cell* 153: 307-319.

REVIEWER 3:

The manuscript by Chatterjee and colleagues investigates the beneficial effects of CSP-TTK21, a small molecule that enhances CBP/p300 histone acetyltransferase activity, in a mouse model of tauopathy. The authors demonstrate that synaptic plasticity and memory deficiencies can be restored along with histone H2B acetylation and specific transcriptional defects. This is a thorough study that presents some novel and relevant findings. Previous studies focused on HDAC inhibitors rather than HAT activators, therefore, the study is potentially interesting and relatively novel. However, the general structure of the manuscript should be improved, and I do not agree with the way most of the bioinformatical analyses were conducted. I also have concerns regarding the reproducibility of the results given the apparent differences across replicates in ChIP-seq experiments.

Main concerns:

1. The abstract and the first paragraph of Discussion introduce the study indicating first that CSP-TTK21 corrects some deficits in a mouse model of tauopathy, and summarizing later the investigation of the genomic events that may underlie this correction. However, surprisingly, in the Results section, the authors describe first the analysis of ChIP-seq and RNA-seq data and only in the last section of Results present the impact of CSP-TTK21 in behavior and physiology (which would indeed provide the rationale for the study). I believe that the manuscript should have the same logical flow outlined in the abstract.

We agree that the manuscript reads much better with an organization that follows the flow of the abstract. We now present the restoration of learning and memory in Figure 1 and that of structural plasticity (dendritic spine formation) and plasticity (LTD) in Figure 2. Genomic effects are presented after, with a first sight at transcriptomics, leading to an evaluation of the rescue and of potential players involved in this rescue (i.e. Klotho and neurotensin), and closing with epigenomic studies that evidence the acetylation status of H2B at different genomic loci in tauopathic tissues and in response to CSP-TTK21 treatment.

2. Figure EV2 shows that the overlap between replicates is very low (between 10-20% of the differential H2Bac peaks). This creates serious concerns regarding the subsequent bioinformatical analyses. SEE POINT 4. The authors should present PCA graphs and Pearson correlation heatmaps (for example) to demonstrate the similarity of their ChIP-seq and RNA-seq replicates and support the robustness of their analyses and the reliability of their conclusions.

3. Most of the bioinformatical analyses seem amateurish and lack statistical rigor. For example, the heatmaps presented in Figure 4, 5 and EV5 are misleading and confusing.

4. The most serious criticisms regarding bioinformatics analyses refer to the abusive use of Venn diagrams. This type of graph can be a useful representation of similarities and divergences, but bears little quantitative and statistical information (they are based on arbitrary thresholds and ignore quantitative information). For example, to define the set of H2B hypoacetylated regions in THY-Tau22 mice, rather than rely all the subsequent analyses in the overlap between two datasets, they could use the information of all the datasets to identify H2Bac-enriched regions through the genome and apply statistical tests to later identify differentially acetylated regions between the two WT and the two THY-Tau22 samples, and between CSP-TTK21 and saline treated mice.

5. The results presented in Figure 3A should illustrate the most important finding in this article: THY-Tau22 and WT mice differ in more than 2500 genes, while CSP-TTK21- and CSP-treated THY-Tau22 only differ in less than 200 genes. However, this partial restoration (less than 10%) seems to be sufficient to ameliorate behavioral, neurological and electrophysiological defects.

Unfortunately, the table format presents partial and biased information. Scatters plots, heat maps, line plots, violin plots or other representations (there are many options) should illustrate this recovery providing at the same time quantitative information and statistical parameters.

In these four points, the reviewer raises the same kind of concerns about the analysis, especially bioinformatics, and their representations. We will try to answer to them sequentially, first for ChIP-seq analyses and second for RNA-seq analyses. But first, we would like to thank the reviewer for these critical comments, as they helped us to improve the manuscript quality.

Generally, we now present a more quantitative analysis of our data. In particular, bioinformatic analyses assessing the reliability of our analyses have been included within the revised version of the manuscript.

ChIP-seq analyses:

The reviewer raised a concern about ChIPseq replicates and the method we have used to analyze these datasets. Importantly, our biological replicates were processed sequentially. Given the cost of ChIP experiments, we have chosen this strategy : to wait and to assess whether a first experiment is working before performing the second one. Specifically, the same experimenter performed the 2 ChIP replicates, but at one year interval, meaning that 1) different lots of reagents (e.g. antibodies, ..) were used between the 2 ChIP replicates, 2) mice were generated at one year interval between the 2 experiments and 3) the production of CSP and CSP-TTK21 molecules (in Dr T. Kundu's laboratory) and treatment were performed at one year interval between the 2 experiments. This can potentially result in some variability. Indeed, PCA analysis of the 2 replicates suggest some batch effect (see Reviewer 3 Figure 1). This is the reason why we did not analyse together the samples, as suggested by the reviewer, but separately. Nevertheless, we are confident in our data and analyses, given that, despite this batch effect, the 2 replicates show same general trends : H2Bac signal is globally decreased at H2Bac-enriched regions in THY-Tau22 mice vs. WT, and globally increased by CSP-TTK21 treatment in THY-Tau22 mice.

We have changed the representation of the epigenetic data presented now in the two last figures (Figures 6, 7 and EV3, 4). The PCA has been added in the revised manuscript (Figure EV3B). H2Bac enrichment at TSS and gene bodies (SeqMiner) are shown for both replicates (Figure 6C). The differential analyses is presented in diagrams as fold changes with adjusted p-values (Fig6D), instead of the venn diagrams. The average of deficits or of rescues are presented with SeqMiner profiles performed on normalized data (Fig 6E, Figure EV3C,D, Figure 7A,B,D,E). H3K27ac SeqMiner profiles that are not differentially regulated are also shown, and support the specificity of differential effect described for H2Bac. Pictures from the genome browser (Figure 7F, Figure EV4C) attest of the quality of the chromatin IPs.

RNA-seq analyses:

We have put a particular attention to include quantitative transcriptomic data in the revised version of the manuscript, with the statistical parameters used.

For differential transcriptomic studies, we show the results in the form of volcano plots for each comparison (Figure 3C,D). The $\log_2(\text{Fold-Change})$ was estimated by DESeq2, with $\text{FDR} < 0.05$ and $|\log_2 \text{ Fold Change}| > 0.2$.

The Heatmaps presenting the clustering of all samples is shown in Reviewer 3 Figure 2.

The reviewer found heatmaps presented in former Figure 4, 5 and EV5 misleading maybe because they were referring to comparisons. These were aimed at evaluating the variations of gene fold changes between two conditions to their variations in the other comparisons, i.e. in figure 4 : are the 98 genes up-regulated by CSP-TTK21 treatment rather increased or decreased in the comparison for learning (Learning vs Home Cage) or in the comparison for pathology (Tau vs WT) ? We thought this was very « visual » as increased genes – appearing in red - were turning green (repressed genes) in the pathology and stayed red in the learning condition. It is true that by doing this, it gives an idea about fold change in the comparisons, but it does not give an estimate of relative expression.

So we have replaced these heatmaps by expression heatmaps (averaged for each group and represented as z-score to normalize the level of expression) for each of the 4 conditions (see Reviewer 3 figure 3). This representation is also very « visual » and shows the clusterization of the different experimental conditions in addition to that of the genes. This representation has also been chosen to show the relative expression of all genes (Figure 3B) in the 4 different conditions presented in Figure 3A in the revised manuscript.

The reviewer argues about a partial restoration as more than 2500 genes are deregulated in TAU VEH vs WT VEH, while CSP-TTK21- and CSP-treated THY-Tau22 only differ in less than 200 genes. This point was also raised by reviewer #1, and here is what we answered :

CSP-TTK21 indeed modifies only a subset of genes relative to all the number of deregulated genes found in CSP-treated mice ($\approx 10\%$). Here again, we checked the rescued learning transcriptome by comparing remaining differential genes in the TAU MOL vs WT VEH conditions. We found it was rescued by about 50% (Figure 3D-G), which could account to improving the phenotype of Thy-Tau22 mice. Mechanistically, this tells us that the beneficial effects of CSP-TTK21 are probably both direct and indirect. This is now discussed in the revised manuscript. We went further into these mechanisms and found that CSP-TTK21 activated targets could be Klotho and Neurotensin, for which we validated an increase in protein expression (Figure 4). There is a growing literature around Klotho, which is documented as exerting beneficial effects in aging, and more recently in a mouse model of AD (Kurosu et al., 2005 ; Leon et al., 2017; Masso et al., 2017; Kuang et al., 2017). The neurotensin system was also found altered in the temporal lobe of AD patients (Gahete et al., 2010) and microinjections of Neurotensin have been shown to restore spatial memory functions and neuronal excitability in APP/PS1 mice (Xiao et al, 2014). Thus, we think that by defining (at least) these specific targets, we have improved the understanding of the mechanisms likely involved in beneficial effect of CSP-TTK21 in THY-Tau22 mice.

We hope that these data providing quantitative informations, statistical parameters and mechanistic explanations are more convincing (Figure 3-5).

6. The description of genomic results and Discussion section could be shortened. I would recommend to restrict these sections to the main finding described in the study: the reversal of transcriptional and histone acetylation deficits by CSP-TTK21, eliminating the abundant digressions of the current version of the manuscript (for instance, the discussion of super-enhancers, a controversial entity, seems completely unnecessary). The manuscript might be better suited to publication as a short report in EMBO Molecular Medicine.

Super-enhancers need to be better characterized in specific cell-types (e.g. it has never been investigated in hippocampal tissues), but we do not think it is a controversial entity as now reported in many publications (original publications : Hnisz et al., 2013 ; Whyte et al., 2013 ; Parker et al., 2013 ; and e.g. new publications and reviews : Pott and Lieb, 2015 ; Niederriter et al., 2015; Vahedi et al., 2015; Yang et al., 2018; Sabari et al., 2018; Shin et al., 2018 to cite a few; and also ours: Achour et al., 2015 ; LeGras et al., 2017). They stand as important regulators and every data that contributes to define them, as our original result on H2Bac co-localization at H3K27ac –enriched regions in the hippocampus, is of interest.

However, the discussion of super-enhancers has been excluded in the revised version of the manuscript. As explained also to reviewer #1, we originally focused on this class of genes because we found a correlation with the fact that these genes were down-regulated upon learning and upon treatment by the molecule. However, we could not draw solid conclusions in terms of mechanistics, and this would indeed require more analyses. In the revised version of the manuscript, we have focused on more obvious changes, that occur at TSS and CBP enhancers (see figures 6E, supplemental 6D and 7A-D).

Taking into account of the amount of original data presented here, we think it is suitable for a regular article in EMBO Molecular Medicine and not for a short report.

7. Similar studies have been conducted with other mouse models of Alzheimer's disease and HDACi rather than HAT activators. It would be very interesting that the article included a direct comparison of the genomic actions of these two families of compounds in these models of disease.

We could not find many studies reporting the effect of HDACi studied at the genomic level in AD mouse models, actually we are aware of only one led in André Fischer's laboratory (Benito et al., 2015). It is cited and commented in the manuscript. It is at this point difficult to compare the epigenomic studies because the targeted histone marks were different (here H2Bac and H3K27ac and in Benito study, H4K12ac). In addition, epigenomic studies in mouse models of Alzheimers are also not very well documented, except that of Tsai's laboratory (Gjoneska et al ; 2014), a study which is also cited in the manuscript.

8. Figure 8 is unnecessary. There is a body of literature relating H3K27ac and enhancers. This evidence does not exist for H2Bac therefore the model is highly speculative and based exclusively in the data presented in this study. What are "Identity genes" and where in the manuscript it is shown that they provide "identity"?

Figure 8 represented a hypothetic model in which we attempted to concile H2Bac and H3K27ac at super-enhancers and their possible role in learning-induced transcriptomic changes ; As discussed above, this is no longer discussed in the manuscript as we do not exploit the super-enhancers data.

Other comments:

9. The results presented in Fig. EV7 are very relevant and should be presented as a main figure.

Thank you for this suggestion. The figure presenting dendritic spine countings in resting mice following a single CSP-TTK21 injection is now presented in figure 2A. It is true that it does show an effect of the molecule in basal conditions, on immature spines, that it is important to underline.

10. The use of the label "H2B4Kac" is a bit confusing next to other labels such as "H2BK5ac". I would recommend to replace it by H2BKw,x,y,z, in which w-z are replaced by the actual lysine positions investigated (that are not detailed anywhere in the manuscript).

We have renamed H2B4Kac with the corresponding lysine numbering : H2BK5K12K15K20. We indeed forgot to list it in the material and method section, it is the reference #07-373 from Millipore. http://www.merckmillipore.com/FR/fr/product/Anti-acetyl-Histone-H2B-Antibody,MM_NF-07-373

11. Some concepts and abbreviations are used without being introduced (e.g., Tau-Ph404, CSP, the use of inflammatory markers should be supported by appropriate references, etc).

We apologize for these mistakes and oversights. We have corrected them throughout the text of the revised manuscript. More particularly :

- Tau-Ph404 is the phosphorylation form of Tau on serine 404. It has been added in Figure 1B legend : « Phosphorylated tau on serine 404 (Tau-Ph404) attest for samples from tauopathic mice.»

- CSP is the name of the molecule used as vehicle to carry TTK21. It is a carbon nanosphere made out of glucose, but it is not an abbreviation. Its first decription is found in Chatterjee et al., 2013 and this reference is cited in the text when appropriate.

- Inflammation in THY-Tau22 mice has been reported several times, and proper references are cited throughout the text, particularly the original one (Schindowski et al., 2006) and a recent one more focused on neuroinflammatory processes in the THY-Tau22 mouse model (Laurent et al., 2017).

12. The "reduced basal neuronal activity" discussed in page 7 correspond to a reduction of the number of cells activated or to decreased expression of IEG in active cells? The interpretation of either result would be different.

We can not answer to this question, which would require to perform thorough stereological analyses of neuronal activity markers, e.g. c-Fos or Egr-1, by immunohistochemistry, in both home cage and in repsonse to learning. It is an interesting question that is ongoing but outside the scope of this study.

13. How was the "predictive motif analysis" referred in page 7 (Fig. 1F) conducted? There is no description in the text.

Predictive motif analyses were performed with GREAT software. It was mentioned in the figure 1 legend (McLean et al., 2010).

<http://bejerano.stanford.edu/great/public/html/index.php>

14. For comparison, the genomic snapshots could also include the profile of CBP from Kim et al. 2010 (Fig. 2D, Fig EV3).

The pictures from genome browser are no longer in the manuscript except that of *Nrp2* (Figure EV4C). However, our ChIP-seq data obtained with H3K27ac were clearer to identify peaks at the gene profile than those obtained with the CBP ChIP from Kim et al (2010) (see the snapshots on Reviewer 3 Figure 4 made for the *c-fos* and *Nrp2* loci as example).

Figure 1 : Principal Component Analyses (PCA). For each condition (WT VEH, TAU VEH, TAU MOL), islands detected with SICER in both replicates were intersected using BEDTools (Quinlan and Hall, 2010) intersect v26. Then, all intersected regions, one dataset per condition, were combined using BEDTools merge v26 to obtain a dataset containing all regions bound in at least one condition. The number of reads falling into the union dataset were computed using BEDTools intersect v26. Data were normalized using the method proposed by Anders and Huber (Anders and Huber, 2010). For PCA plotting, data were transformed in order to stabilize variance using the DESeq2 07/09/18 10:26Bioconductor package (Love et al., 2014).). PCA analyses show that intrinsic variability of rep1 is higher than that of rep2 (A). However, the samples appear clustered according to group (e.g. TAU VEH, WT VEH, TAU MOL).

RNA-seq samples (Figure 1) Resting mice

RNA-seq samples (Figure 3) Learning mice / Molecule treatment

Figure 2: Heatmaps with a clustering of all samples for both RNA-seq studies. The hierarchical clustering was calculated using the UPGMA (unweighted pair group method with arithmetic means) with the SERE coefficient as the distance measured.

A

B

Figure 3: Heatmaps presented in the former and the revised versions of the manuscript . **A** The 98 up-regulated genes in the comparison TAU MOL vs TAU VEH (Left) in the former version, presented as fold change in the different comparisons and (Right) in the revised version, presented as relative expression (z-score) of the average of samples in each conditions (Figure 4B). Clustering was performed using the Unweighted Pair Group Method with Arithmetic mean method and the Pearson's distance.

B Same representations for the 82 down-regulated genes in the comparison TAU MOL vs TAU VEH, presented in the revised manuscript in figure 5B.

Figure 4: Snapshots at the genome Browser showing published CBP profiles (Kim et al., 2010) with Abcam and Millipore antibodies (in samples treated with KCl, first and second from top or untreated (un), third and fourth from top) at the *c-fos* and the *Nrp2* loci, together with our H3K27ac ChIP-seq data.

References

- Achour M, Le Gras S, Keime C, Parmentier F, Lejeune FX, Boutillier AL, Neri C, Davidson I, Merienne K (2015) Neuronal identity genes regulated by super-enhancers are preferentially down-regulated in the striatum of Huntington's disease mice. *Human molecular genetics* 24: 3481-3496
- Anders S, Huber W (2010) Differential expression analysis for sequence count data. *Genome biology* 11: R106
- Hnisz D, Abraham BJ, Lee TI, Lau A, Saint-Andre V, Sigova AA, Hoke HA, Young RA (2013) Super-enhancers in the control of cell identity and disease. *Cell* 155: 934-947
- Laurent C, Dorothée G, Hunot S, Martin E, Monnet Y, Duchamp M, Dong Y, Légeron FP, Leboucher A, Burnouf S, Faivre E, Carvalho K, Cailliez R, Zommer N, Demeyer D, Jouy N, Sazdovitch V, Schraen-Maschke S, Delarasse C, Buée L, Blum D. Hippocampal T cell infiltration promotes neuroinflammation and cognitive decline in a mouse model of tauopathy. *Brain*. 2017 Jan;140(1):184-200.
- Le Gras S, Keime C, Anthony A, Lotz C, De Longprez L, Brouillet E, Cassel JC, Boutillier AL, Merienne K (2017) Altered enhancer transcription underlies Huntington's disease striatal transcriptional signature. *Scientific reports* 7: 42875
- Love MI, Huber W, Anders S (2014) Moderated estimation of fold change and dispersion for RNA-seq data with DESeq2. *Genome biology* 15: 550
- McLean CY, Bristor D, Hiller M, Clarke SL, Schaar BT, Lowe CB, Wenger AM, Bejerano G (2010) GREAT improves functional interpretation of cis-regulatory regions. *Nature biotechnology* 28: 495-501
- Niederriter AR, Varshney A, Parker SC, Martin DM. Super Enhancers in Cancers, Complex Disease, and Developmental Disorders. *Genes (Basel)*. 2015 Nov 9;6(4):1183-200.
- Parker SC, Stitzel ML, Taylor DL, Orozco JM, Erdos MR, Akiyama JA, van Bueren KL, Chines PS, Narisu N; NISC Comparative Sequencing Program, Black BL, Visel A, Pennacchio LA, Collins FS; National Institutes of Health Intramural Sequencing Center Comparative Sequencing Program Authors; NISC Comparative Sequencing Program Authors. Chromatin stretch enhancer states drive cell-specific gene regulation and harbor human disease risk variants. *Proc Natl Acad Sci U S A*. 2013 Oct 29;110(44):17921-6.
- Pott S, Lieb JD. What are super-enhancers? *Nat Genet*. 2015 Jan;47(1):8-12.

Quinlan, A. R. & Hall, I. M. (2010) BEDTools: a flexible suite of utilities for comparing genomic features. *Bioinformatics* 26: 841–842.

Sabari BR, Dall'Agnese A, Boija A, Klein IA, Coffey EL, Shrinivas K, Abraham BJ, Hannett NM, Zamudio AV, Manteiga JC, Li CH, Guo YE, Day DS, Schuijers J, Vasile E, Malik S, Hnisz D, Lee TI, Cisse II, Roeder RG, Sharp PA, Chakraborty AK, Young RA. Coactivator condensation at super-enhancers links phase separation and gene control. *Science*. 2018 Jun 21.

Schindowski K, Bretteville A, Leroy K, Bégard S, Brion JP, Hamdane M, Buée L. Alzheimer's disease-like tau neuropathology leads to memory deficits and loss of functional synapses in a novel mutated tau transgenic mouse without any motor deficits. *Am J Pathol*. 2006 Aug;169(2):599-616.

Shin HY. Targeting Super-Enhancers for Disease Treatment and Diagnosis. *Mol Cells*. 2018 Jun;41(6):506-514.

Vahedi G, Kanno Y, Furumoto Y, Jiang K, Parker SC, Erdos MR, Davis SR, Roychoudhuri R, Restifo NP, Gadina M, Tang Z, Ruan Y, Collins FS, Sartorelli V, O'Shea JJ. Super-enhancers delineate disease-associated regulatory nodes in T cells. *Nature*. 2015 Apr 23;520(7548):558-62.

Whyte WA, Orlando DA, Hnisz D, Abraham BJ, Lin CY, Kagey MH, Rahl PB, Lee TI, Young RA (2013) Master transcription factors and mediator establish super-enhancers at key cell identity genes. *Cell* 153: 307-319.

Yang R, Wu Y, Ming Y, Xu Y, Wang S, Shen J, Wang C, Chen X, Wang Y, Mao R, Fan Y. A super-enhancer maintains homeostatic expression of Regnase-1. *Gene*. 2018 Aug 30;669:35-41.

2nd Editorial Decision

13 August 2018

Thank you for the submission of your revised manuscript to EMBO Molecular Medicine. We have now received the enclosed reports from the referees that were asked to re-assess it. As you will see the reviewers are now globally supportive and I am pleased to inform you that we will be able to accept your manuscript pending final amendments.

***** Reviewer's comments *****

Referee #1 (Remarks for Author):

Overall, the authors have addressed my comments. This reviewer is still not fully convinced that the exact mechanism of drug action has been revealed, but the therapeutic effect is convincing enough to warrant publication. The data to the reviewers (not included in the current manuscript) has significantly influenced my decision. In order for the reader to come to the same conclusions all data presented to the reviewer should be included in the manuscript.

Referee #2 (Remarks for Author):

The authors have taken the critiques seriously and try as much as they could to improve their manuscript based on the comments. They have addressed my previous comments. This paper is very interesting and should be published in EMBO Mol Med.

Referee #3 (Remarks for Author):

The revised manuscript presents substantial improvements. The flow of the paper has improved, and the description and analysis of genomic data is also much better than in the original submission although there is still room for improvement. Overall, the authors addressed all my main concerns in their rebuttal letter. However, the authors should revise figures and legends because some descriptions are misleading and relevant information is missing or unclear. For example:

- The section header "Effect of CSP-TTK21 on the learning-induced transcriptome of THY-Tau22 mice" is inaccurate. The experimental design outlined in Fig. 3A does not allow to investigate this

because the authors did not tackle the "the learning-induced transcriptome of THY-Tau22 mice". To approach that question, they could however investigate the behavior of the "learning-induced transcriptome" of WT mice - as defined in the middle panel of Fig. 3C - in THY-Tau22 mice. However, I believe they did not conduct such analysis.

- Figure 3C,D - I can guess what is the meaning of the numbers in the upper left and right corners but this should be explained in the legend. Also, the number in the X and Y-axes are too small.
- Figure 3B should be introduced after Figure 3C (particularly after the right panel of Figure 3C that shows the original analysis that led to the gene set presented in panel B)
- Figure 3F - Does "300 mostly deregulated genes..." means "300 most deregulated genes"?
- Figure 4B and 5B - I believe that the description of these figure is wrong. The authors indicate "heatmap representing expression z-score of the 98/82 significantly upregulated/downregulated genes in all experimental conditions". However, they are referring to the 98/82 genes that are upregulated/downregulated in the TAU MOL vs TAU VEH comparison (and then, they present the values in all the conditions). Also, the gene symbols in the right columns are impossible to read and should be therefore eliminated.
- Figure EV1B: This panel is not a volcano plot. Also, the legends in both axes are impossible to read.
- Figure EV4A-B: The color code of the graph is not shown.
- Figure 2 in the rebuttal to my comments: Either the vertical or the horizontal labels in the two heatmaps are wrong. The results, as presented, are not possible.

2nd Revision - authors' response

31 August 2018

Reviewer's comments

We would like to thank the three referees for their insightful and sound comments that certainly contributed to a great improvement of the paper quality.

Referee #1 (Remarks for Author):

Overall, the authors have addressed my comments. This reviewer is still not fully convinced that the exact mechanism of drug action has been revealed, but the therapeutic effect is convincing enough to warrant publication. The data to the reviewers (not included in the current manuscript) has significantly influenced my decision. In order for the reader to come to the same conclusions all data presented to the reviewer should be included in the manuscript.

A new Supplemental Figure S1 (appearing in the appendix) has been done to provide the readers with the RNA-seq validation data from both experiments (acute injections in behaving mice and chronic injections in resting mice) that were provided to reviewer 1.

It appears in the text of the manuscript as this (p.11) :

« In line, CSP-TTK21 induced the expression of a series of IEGs (e.g., *Arc*, *c-fos*, and *egr1*) in both basal and learning conditions (Fig 4D; Appendix Supplemental Figure S1D,E), suggesting that the molecule improved neuronal activity. »

Referee #2 (Remarks for Author):

The authors have taken the critiques seriously and try as much as they could to improve their manuscript based on the comments. They have addressed my previous comments. This paper is very interesting and should be published in EMBO Mol Med.

Referee #3 (Remarks for Author) :

The revised manuscript presents substantial improvements. The flow of the paper has improved, and the description and analysis of genomic data is also much better than in the original submission although there is still room for improvement. Overall, the authors addressed all my main concerns in their rebuttal letter. However, the authors should revise figures and legends because some descriptions are misleading and relevant information is missing or unclear. For example:

- The section header "Effect of CSP-TTK21 on the learning-induced transcriptome of THY-Tau22 mice" is inaccurate. The experimental design outlined in Fig. 3A does not allow to investigate this because the authors did not tackle the "the learning-induced transcriptome of THY-Tau22 mice". To approach that question, they could however investigate the behavior of the "learning-induced transcriptome" of WT mice - as defined in the middle panel of Fig. 3C - in THY-Tau22 mice. However, I believe they did not conduct such analysis.

This is right, we did not include Home Cage THY-Tau22 mice in this series of experiments, so we couldn't evaluate the effect of learning *per se* in the TAU line. Our section header is inaccurate in that our study is not restricted to the learning-induced transcriptome but includes all TAU-deregulated genes during a learning task, so we have replaced this section header by « Transcriptomic effects of CSP-TTK21 in learning THY-Tau22 mice ». This has also been changed at several places in the manuscript where « the learning-induced transcriptome » appeared. This was replaced by « the transcriptome induced in learning mice » (p.3 ; p.11 ; p.15 ; p.16 ; p.56). The independent analyses of the « learning-induced transcriptome » of WT mice (deregulated genes in WT VEH vs WT VEH_HC) in the THY-Tau22 mice could have more particularly investigated this point, but we already presented many comparisons in the manuscript and rather wanted to focus on the genes directly regulated by CSP-TTK21 (Fig 4 and 5).

- Figure 3C,D -I can guess what is the meaning of the numbers in the upper left and right corners but this should be explained in the legend. Also, the number in the X and Y-axes are too small.

This has been added and the size of axe legends has been increased.

- Figure 3B should be introduced after Figure 3C (particularly after the right panel of Figure 3C that shows the original analysis that led to the gene set presented in panel B)

Thank you for this suggestion that improves the reading of these results. Figure 3B represents the gene expression levels (as row z-score) of the deregulated genes found in the TAU VEH vs WT VEH comparison (2756 total) for all experimental conditions. So this corresponds to the left panel of former Figure 3C. This has been changed so that indeed, we can see the volcano plot resulting from the differential analyses of that condition (Figure 3B) and then the expression of these genes in all the experimental conditions (Figure 3C). The rest of the figure has been rebalanced and the 300 most deregulated genes are now shown in supplemental Figure EV2A. Figure legends have been rewritten accordingly.

These changes appear on pages 9-10 of the new version of the manuscript « In response to learning, we found that 2756 genes were .../... the transgene expression (Fig EV2D). ».

- Figure 3F - Does "300 mostly deregulated genes..." means "300 most deregulated genes"?

Yes, sorry for this english grammar mistake. This is changed on the figure (now EV2A) and in the text.

- Figure 4B and 5B - I believe that the description of these figure is wrong. The authors indicate "heatmap representing expression z-score of the 98/82 significantly upregulated/downregulated genes in all experimental conditions". However, they are referring to the 98/82 genes that are upregulated/downregulated in the TAU MOL vs TAU VEH comparison (and then, they present the values in all the conditions). Also, the gene symbols in the right columns are impossible to read and should be therefore eliminated.

We apologize for this description of the figure that is indeed confused. This paragraph has been rephrased (p.10) and figure legends 4 and 5 have been rephrased (p.57,58). It is true that on the complete pdf downloaded from the EMM site, enlarging the figure does not allow to read the gene symbols, but the original high quality figure we provided do allow to read them. So we have chosen to leave these names as we will upload high quality figures and this should give the opportunity to the reader to focus on these names if desired.

- Figure EV1B: This panel is not a volcano plot. Also, the legends in both axes are impossible to read.

This is a scatter plot. Axis legends have been enlarged.

- Figure EV4A-B: The color code of the graph is not shown.

This has been added.

- Figure 2 in the rebuttal to my comments: Either the vertical or the horizontal labels in the two heatmaps are wrong. The results, as presented, are not possible.

We apologize for this, we have made labeling errors when reformatting the figures because the groups from the original report were named differently from that nomenclature used in the manuscript (e.g. TAU was labeled AD, MOL was labeled TTK). This is now corrected (below).

RNA-seq samples (Figure 1)
Resting mice

RNA-seq samples (Figure 3)
Learning mice / Molecule treatment

Corresponding Author Name: Anne-laurence BOUTILLIER

Manuscript Number: EMM-2017-08587-T